# Autapses enhance bursting and coincidence detection in neocortical pyramidal cells

Luping Yin [1,2], Rui Zheng[1,2], Wei Ke[1], Quansheng He[1], Yi Zhang[1], Junlong Li[1], Bo Wang [1,2], Zhen Mi[1], Yue-sheng Long[3], Malte J. Rasch [1], Tianfu Li[4], Guoming Luan[5] & Yousheng Shu [1]

Autapses are synaptic contacts of a neuron's axon onto its own dendrite and soma. In the neocortex, self-inhibiting autapses in GABAergic interneurons are abundant in number and play critical roles in regulating spike precision and network activity. Here we examine whether the principal glutamatergic pyramidal cells (PCs) also form functional autapses. In patch-clamp recording from both rodent and human PCs, we isolated autaptic responses and found that these occur predominantly in layer-5 PCs projecting to subcortical regions, with very few in those projecting to contralateral prefrontal cortex and layer 2/3 PCs. Moreover, PC autapses persist during development into adulthood. Surprisingly, they produce giant post-synaptic responses (∼5 fold greater than recurrent PC-PC synapses) that are exclusively mediated by AMPA receptors. Upon activation, autapses enhance burst firing, neuronal responsiveness and coincidence detection of synaptic inputs. These findings indicate that PC autapses are functional and represent an important circuit element in the neocortex.

[1] State Key Laboratory of Cognitive Neuroscience and Learning & IDG/McGovern Institute for Brain Research, Beijing Normal University, 19 Xinjiekou Wai Street, Beijing 100875, China. [2] Institute of Neuroscience and State Key Laboratory of Neuroscience, Chinese Academy of Sciences, University of Chinese Academy of Sciences, 320 Yueyang Road, Shanghai 200031, China. [3] Institute of Neuroscience and the Second Affiliated Hospital of Guangzhou Medical University, Guangzhou 501260, China. [4] Department of Neurology, Epilepsy Center, Sanbo Brain Hospital, Capital Medical University, Xiangshan Yikesong 50, Beijing 100093, China. [5] Department of Neurosurgery, Epilepsy Center, Sanbo Brain Hospital, Capital Medical University, Xiangshan Yikesong 50, Beijing 100093, China. These authors contributed equally: Luping Yin, Rui Zheng, Wei Ke, Quansheng He. Correspondence and requests for materials should be addressed to Y.S. (email: yousheng@bnu.edu.cn)

Autapses are special synaptic contacts formed in a single neuron between its own axon and dendrites (or soma). They were so named by van der Loos and Glaser to describe self-innervating synapses in Golgi-stained neocortical pyramidal cells (PCs)[1]. Later anatomical observations found putative autaptic contacts in different types of neurons in the central nervous system, including striatum medium spiny neurons[2–4] and substantial nigra pars reticulata neurons[5]. Importantly, electron microscopy studies revealed the typical synaptic organization at autaptic contacts of neocortical PCs[6], neocortical, and hippocampal inhibitory basket cells[7–9]. Electrophysiological recordings from inhibitory interneurons in cortical[10–13] and cerebellar slices[14] showed the occurrence of GABA_A receptor-mediated autaptic currents. In addition, activation of these GABAergic autapses inhibits subsequent action potential generation[11] and regulates spike precision in neocortical fast-spiking interneurons[10], indicating that autapses of inhibitory interneurons are functional structures and play important roles in regulating neuronal signaling.

Prior anatomical analysis of the abundance of autapses in PCs resulted in conflicting findings. In agreement with the original observation by van der Loos and Glaser that half of the examined PCs in rabbit occipital cortex possessed autapses[1], a large proportion (~80%) of layer-5 PCs in somatosensory cortex of juvenile rats showed autaptic connections[6]. In contrast, much rarer autapses were found in PCs in other cortical layers (2–4) of adult cat visual cortex[8]. These inconsistent observations could be attributable to differences in species, age, brain region, and cortical layer. It could be possible that autapses may selectively occur in certain types of PCs, similar to GABAergic autapses that selectively form in fast-spiking cells[8,11]. Although excitatory autaptic currents have been recorded in cultured hippocampal neurons with extraordinarily abundant autapses[15–17], no physiological evidence so far demonstrates whether PC autapses are functional in cortical tissue. Previous experiments have tried to detect autaptic responses in layer-5 PCs in slices[6], but were unsuccessful possibly due to masking of fast autaptic currents by broad AP waveforms.

The functional significance of PC autapses, if present, remains unclear. van der Loos and Glaser[1] proposed a gating hypothesis that autapses may provide self-inhibition by electrically shunting and gating synaptic inputs arrived at a dendritic location distal to the autaptic contacts. Modeling studies predicted roles of excitatory autapses in generating persistent activity associated with short-term memory[18] and regulating neuronal signaling[19], although positive feedback may destabilize the network activity. An experimental study in *Aplysia* indeed found that muscarinic autaptic excitation could cause prolonged depolarization and persistent activity in motor neurons associated with feeding behavior[20,21].

In this study, we sought to examine whether neocortical PCs form functional autaptic connections and investigate their functional significance in signal processing. Using a combination of dual soma-axon recording and 2-photon laser axotomy or recording in the presence of $Sr^{2+}$, which desynchronizes neurotransmitter release, in PCs of mouse prefrontal cortex (PFC) and human frontal lobe, we found that autapses selectively occur in layer-5 PCs projecting to subcortical brain regions and mediate glutamatergic synaptic transmission. In addition, we also revealed a functional role of autapses in promoting neuronal responsiveness, burst firing and coincidence detection in these neocortical principal cells.

## Results

**Neocortical layer-5 PCs form autapses.** To isolate autaptic responses in PCs, we first made simultaneous whole-cell recordings from both the soma and the axon bleb (i.e., cut end of the axon formed during slicing) of single layer-5 PCs in acute slices of mouse prefrontal cortex[22] (postnatal day 13–21, see Methods). Next, we cut the axon with a two-photon laser close to the soma and just beyond the axon initial segment (Fig. 1a). Before axotomy, stimulation of the axon (0.7–1.0 ms in duration, 1–2 Hz) evoked action currents at the soma ($V_{hold}$: −70 mV), resulting from the arrival of backpropagating APs invading the soma. After axotomy, somatic action currents were blocked (Fig. 1b) and close examination of the residual currents revealed synaptic current-like events (Fig. 1c). These events had an average peak amplitude of −149 ± 11 pA (mean ± SEM, $n = 62$ cells). They could be fully blocked by bath application of the AMPA/kainate receptor antagonist CNQX (20 μM, $n = 5$ cells, Fig. 1d), indicating that they were autaptic excitatory postsynaptic currents (aEPSCs). Because the background spontaneous EPSCs showed no significant change in the peak amplitude, the rise time and the decay time constant before and after the axotomy (Supplementary Figure 1), aEPSCs should reflect their actual waveform before axotomy. Switching the somatic recording to current-clamp mode revealed large autaptic excitatory postsynaptic potentials (aEPSPs, Fig. 1c). The small current deflection preceding the autaptic current reflected the dramatically reduced action currents following axotomy. The short onset latencies of these aEPSCs (1.94 ± 0.06 ms, $n = 45$ cells; Fig. 1e) agree well with monosynaptic transmission[23].

We further confirmed the occurrence of monosynaptic aEPSCs by using $Sr^{2+}$-containing bath solution (8 mM $SrCl_2$) to desynchronize and thus postpone neurotransmitter release[24]. We made simultaneous recordings from two closely neighboring PCs (<50 μm apart), and stimulated only one of them with trains of voltage pulses (4 pulses at 20 Hz every 20 s). These pulses not only gave rise to action currents in the stimulated cell, but also to numerous asynchronous synaptic events, but in the stimulated cell only ($n = 6$ pairs, Fig. 2a). These desynchronized events had an average peak amplitude of −19.3 ± 2.6 pA and an average half-width of 4.31 ± 0.38 ms (rise time: 0.91 ± 0.03 ms; decay time: 3.92 ± 0.40 ms). Since the neighboring recorded cell did not show synaptic events, this suggests that these events resulted from the activation of autapses rather than the local network. Bath application of kynurenic acid (Kyn, 1.5 mM, $n = 6$ cells), an ionotropic glutamate receptor blocker, diminished these delayed synaptic events (Fig. 2b, c). Including a fast chelator of $Ca^{2+}$ and $Sr^{2+}$, BAPTA (5 mM), in the recording pipette solution resulted in a progressive reduction of desynchronized autaptic events following establishing the whole-cell recording mode ($n = 6$ cells; Fig. 2d). These results are consistent with a calcium-dependent mechanism and support the conclusion that synaptic glutamate was released from the recorded cell.

**Selective formation of autapses in neocortical PCs.** To resolve the mismatch of anatomical findings on the abundance of autapses in PCs, we tested whether the occurrence of autapses was dependent on cortical layer or projection targets of PCs. The original anatomical findings suggested that autapses could be abundant (50%[1], 80%[6]) or rare (10%[8]). In our experiments, we randomly recorded from layer-5 PCs (maximum length of soma: 23.6 ± 0.8 μm; width: 18.0 ± 0.7 μm, $n = 33$ cells) in $Sr^{2+}$-ACSF and revealed that a large proportion of these cells possessed autapses (56.4%, $n = 163$ cells), similar to the observation in laser axotomy experiments (55.2%, $n = 29$ cells, Fig. 2e). These values are actually underestimated because slicing the brain cuts and reduces the complexity of dendrites and axons. Surprisingly, much fewer layer 2/3 PCs (11.5%, $n = 61$ cells, Fig. 2e) showed autaptic currents, suggesting a layer-specific formation of

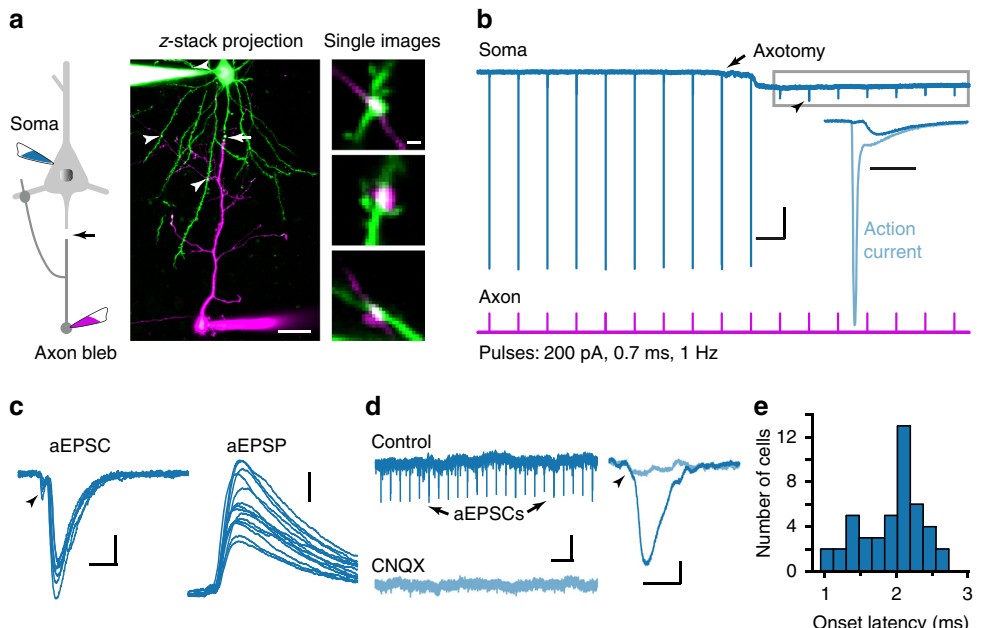

**Fig. 1** Layer-5 PCs form functional autapses. **a** Left, schematic drawing of dual soma and axon-bleb recording with two-photon laser axotomy (black arrow) just beyond the axon initial segment. Middle, an example image showing dual recording and axotomy (white arrow) in a layer-5 PC (scale bar: 50 μm). Patch pipettes for the soma and the axon were filled with internal solutions containing Alexa Fluo-488 (green) and -594 (magenta) fluorescent dyes, respectively. Due to the block of dye diffusion after axotomy, the two dyes accumulated in somatodendritic and axonal compartments, respectively. Arrowheads indicate putative autaptic contacts. Right, higher magnification of the contacts (scale bar: 2 μm). **b** A representative recording. Axon stimulation evoked action currents at the soma ($V_{hold}$: −70 mV). After axotomy (arrow), AP backpropagation was blocked as indicated by the absence of somatic action currents. Inset, overlay of an action current and a residual current (arrowhead). Scale bars: 1 s/1 nA (inset, 5 ms). **c** Overlay of somatic responses to axon stimulation in voltage-clamp mode (left, current traces from the gray box in **b**) and current-clamp mode (right). Note the occurrence of aEPSCs and aEPSPs. The initial current deflections (arrowhead) indicate the arrival of backpropagating APs through the incomplete axotomy site. Scale bars: 5 ms/100 pA (2 mV for aEPSP). **d** CNQX (10 μM) blocked the occurrence of aEPSCs (scale bars: 2 s/50 pA). Expanded traces (right) before and after CNQX application are shown (scale bars: 10 ms/10 pA). **e** Distribution of the onset latencies ($n = 45$ cells)

autapses in neocortical PCs. Similar to those in layer-2/3, a subpopulation of layer-5 PCs also projects to contralateral PFC (cPFC). Interestingly, only 2.6% ($n = 78$ cells) of layer-5 PCs labeled by retrograde beads injection to cPFC formed autapses. In sharp contrast, PCs labeled by retrograde beads injected in pontine nuclei and habenula possessed far more autapses (59.3%, $n = 86$ cells and 84.8%, $n = 66$ cells, respectively; Fig. 2e and Supplementary Figure 2), indicating a selective formation of autapses in certain types of PCs. These autapses were not transient developmental structures, since 29.3% ($n = 82$ cells) of PCs of adult mice (8 weeks old) had aEPSCs. To test whether PC autapses existed also in adult human cortex, we recorded from human layer-5 PCs and tested for functional autapses. In 28.6% cells ($n = 28$ cells from two adult epilepsy patients) recorded from slices of human frontal cortex in $Sr^{2+}$-ACSF (see Methods), we detected CNQX-sensitive autaptic currents (Fig. 2e, f and Supplementary Figure 3). The average peak amplitude of these desynchronized human aEPSCs was −15.3 ± 1.3 pA and the half-width was 5.62 ± 0.15 ms (rise time: 1.09 ± 0.02 ms; decay time constant: 6.35 ± 0.32 ms). These findings show that excitatory autapses made by PCs are evolutionarily conserved in mouse and human cortex and persist during brain development into adulthood.

**Autaptic synapses are much stronger than PC–PC synapses.** Next, we compared the properties of unitary aEPSCs evoked after axotomy with those of EPSCs between two neighboring PCs (Fig. 3a). We found that aEPSCs were large with an average peak amplitude of −149 ± 11 pA across cell population ($n = 62$ cells,

the average for individual cells ranged from −27 to −432 pA, single trials could reach up to −524 pA), 5.2 fold greater than that of PC–PC EPSCs (−28.6 ± 4.9 pA, $n = 10$ pairs, range from −16.3 to −68.3 pA, $Z = -4.78$, $P = 1.8 \times 10^{-6}$, Wilcoxon rank-sum test; Figure 3a, b). The CV of aEPSCs was significantly lower than that of PC–PC EPSCs (0.25 ± 0.01, $n = 13$ cells vs. 0.37 ± 0.01, $n = 10$ pairs, $t_{21} = -4.86$, $P = 8.3 \times 10^{-5}$, two sample Student's $t$-test). The aEPSCs possessed lower failure rates than PC–PC EPSCs (0.03 ± 0.01, $n = 13$ cells vs. 0.11 ± 0.02, $n = 10$ pairs, $Z = -2.55$, $P = 0.01$, Wilcoxon rank-sum test; Figure 3c), and showed slightly longer onset latency (1.94 ± 0.05 ms, $n = 45$ cells vs. 1.61 ± 0.08 ms, $n = 10$ pairs, $t_{53} = 2.41$, $P = 0.02$, two sample Student's $t$-test; Fig. 3d). No significant differences were found in the rise time (0.99 ± 0.02 ms, $n = 57$ cells vs. 0.85 ± 0.05 ms, $n = 10$ pairs, $Z = 1.84$, $P = 0.07$) and the decay time constant (4.63 ± 0.11 ms vs. 5.23 ± 0.50 ms, $Z = -0.99$, $P = 0.32$, Wilcoxon rank-sum test for both; Fig. 3e).

We next examined the short-term plasticity at autapses by delivering trains of stimulation at 10, 20, and 50 Hz every 10 s through the axonal recording pipette. The amplitude of aEPSCs reduced exponentially with time constants of 55.2 ± 19.1, 64.2 ± 5.2, and 47.0 ± 7.7 ms ($n = 7$ cells; Fig. 3f, g) for those particular frequencies, respectively. PC–PC EPSCs, however, showed initial facilitation and then depression (Fig. 3g), which was in agreement with previous findings in ferret prefrontal cortex[25]. The extent of depression was similar between aEPSCs and PC–PC EPSCs, the amplitude ratios of 8th to 1st PSCs at 20 Hz were 0.47 ± 0.05 ($n = 7$ cells) and 0.60 ± 0.05 ($n = 10$ pairs, $Z = -1.70$, $P = 0.09$, Wilcoxon rank-sum test), respectively.

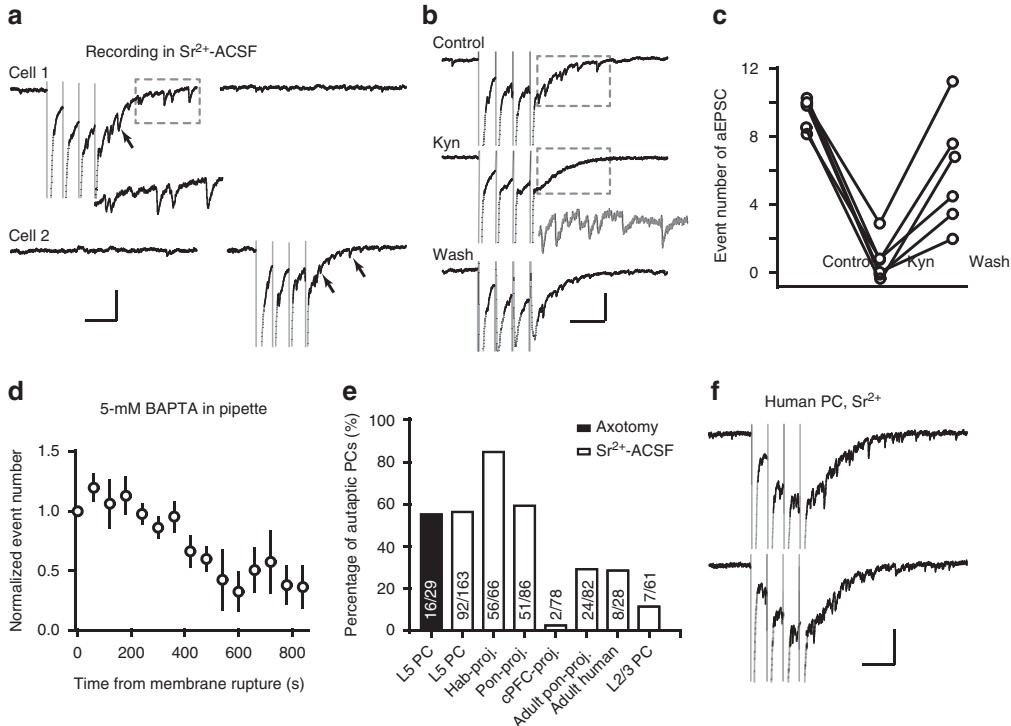

**Fig. 2** Selective formation of autapses in neocortical PCs. **a** Simultaneous recording from two neighboring PCs (<50 µm apart) in Sr²⁺-ACSF. Action currents (truncated, gray) were followed by delayed asynchronous autaptic currents (arrow). No obvious increase in synaptic events was detected in the unstimulated PC. Inset, an enlarged trace of the boxed area (x2). Scale bars: 100 ms/50 pA. **b** Bath application of 1.5 mM Kyn blocked the desynchronized aEPSCs. Inset, a result trace from subtraction of the boxed regions (Control—Kyn). Scale bars: 100 ms/50 pA (50 ms/25 pA for inset). **c** Group data showing the effect of Kyn ($n = 6$ cells). **d** Inclusion of BAPTA (5 mM) in the pipette solution progressively blocked the asynchronous aEPSCs. Data are represented as mean ± SEM. **e** Percentage of autaptic cells in layer-5 and layer 2/3 PCs, and those projecting to the contralateral PFC (cPFC-proj.) or subcortically to pons (Pon-proj.) and habenula (Hab-proj.). For injection of retrograde beads at pons and habenula, see Supplementary Figure 2. **f** Example traces from a layer-5 PC in human cortical slice (scale bars: 100 ms/100 pA). Also see Supplementary Figure 3

The paired-pulse ratio at 20 Hz (aEPSC₂/aEPSC₁: 0.74 ± 0.06, $n = 7$ cells) was less than 1 and significantly lower than that of PC–PC EPSCs (1.11 ± 0.08, $n = 10$ pairs, $Z = -2.68$, $P = 0.005$, Wilcoxon rank-sum test; Fig. 3h), suggesting a higher probability of release ($p$) at autapses. Indeed, fitting the train responses with a model of dynamic synaptic transmission[25,26] revealed that the $p$ at autapses is significantly higher than that at PC–PC synapses (0.53 ± 0.03, $n = 7$ cells vs. 0.32 ± 0.03, $n = 10$ pairs, $t_{15} = 4.81$, $P = 0.0002$, two sample Student's $t$-test; Fig. 3i).

Three-dimensional reconstruction[27] of the randomly recorded layer-5 PCs (without axotomy) in Sr²⁺-ACSF showed that, on average, the number of putative autapses per cell was 2.80 ± 0.59 ($n = 10$ cells; see Methods) in PCs with functional autaptic connections, significantly more than those showing no autaptic currents (0.56 ± 0.24, $n = 9$ cells, $t_{17} = 3.2$, $P = 0.0052$, two sample Student's $t$-test). The autaptic contacts per cell were approximately three times those of connected PC–PC pairs (1.14 ± 0.14, $n = 7$ pairs). Most of the putative autapses formed at the basal dendrites ($n = 24/28$ contacts) (Fig. 4a, b). The dendritic and axonal distances to the soma were 72.3 ± 11.4 and 239 ± 16 µm, respectively (Fig. 4b, c).

Together, these results revealed that, although autapses and PC–PC synapses share similar postsynaptic response kinetics, autapses produce much larger postsynaptic responses than PC–PC synapses. The generation of giant aEPSCs could be attributable to the initial higher probability of release and more autaptic contacts.

**Autaptic currents contain no NMDA component.** Recurrent excitatory synapses between cortical PCs express NMDA receptors[23,28,29], we next examined whether autaptic responses also contain an NMDA component. We modified the Sr²⁺-ACSF by omitting Mg²⁺ but adding 10 µM glycine (or 100 µM D-serine) to enhance NMDA current at negative $V_m$ levels and blocked APs, GABA synapses, and potassium channels (0.5 µM tetrodotoxin (TTX) and 50 µM picrotoxin (PTX) and Cs⁺-based pipette solution; Fig. 5a). Somatic voltage pulses (10–1000 ms, 100–200 mV) evoked asynchronous synaptic events in the recorded PCs (Fig. 5b). Since AP generation was blocked by TTX, these events actually reflected monosynaptic transmission at autapses. The average peak amplitude of the desynchronized aEPSCs was −8.49 ± 2.56 pA ($n = 5$ cells), similar to that of PC–PC EPSCs recorded in the same conditions (−7.96 ± 2.68 pA, $n = 5$ pairs, $t_8 = 0.14$, $P = 0.89$, two sample Student's $t$-test).

Surprisingly, bath application of NBQX (10 µM) could completely block the autaptic events. After drug application, the current noise levels before and after the voltage pulse showed no significant difference (1.49 ± 0.21 vs. 1.61 ± 0.17 pA, $n = 6$ cells, $t_5 = -1.56$, $P = 0.18$, paired Student's $t$-test; Fig. 5b, c), suggesting no contribution of NMDA receptors to autaptic responses. Consistent with the presence of NMDA receptors in recurrent excitatory synapses, desynchronized EPSCs evoked by extracellular stimulation in the presence of Sr²⁺ (but without TTX) showed a reduction in the decay time constant after the application of 50 µM APV, an NMDA receptor blocker (from

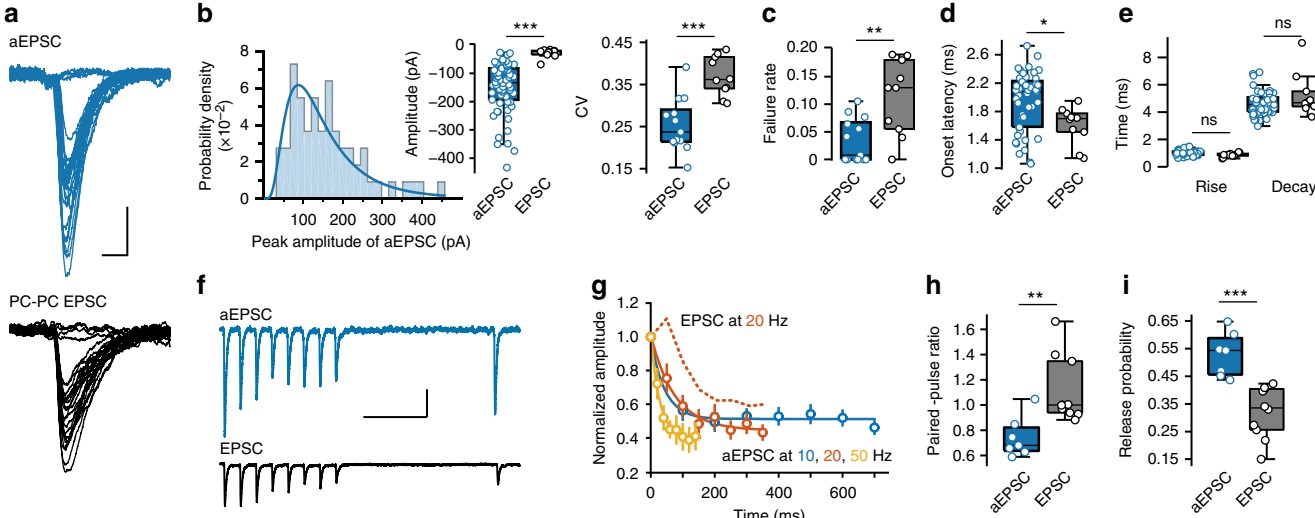

**Fig. 3** Properties of unitary aEPSCs and PC–PC EPSCs. **a** Two representative recordings showing aEPSCs (top, simultaneous soma and axon recording with laser axotomy, overlap of 26 trials) and PC–PC EPSCs (bottom, paired PC–PC recording, 25 trials). Scale bars: 5 ms/100 pA for aEPSC and 20 pA for EPSC. **b** Distribution of aEPSC peak amplitudes with a lognormal fit (left). Middle and right, comparisons of the average peak amplitude (Wilcoxon rank-sum test) and CV (two sample Student's $t$-test) of aEPSC and EPSC. Boxplots represent the median and interquartile range and whiskers represent 1.5× interquartile range. **c**, **d**, **e** Group data showing the average failure rates (Wilcoxon rank-sum test), the onset latency (two sample Student's $t$-test), the rise time and decay time constant (Wilcoxon rank-sum test) of aEPSC and EPSC. **f** Top, an average trace showing short-term plasticity of aEPSCs in response to 20-Hz stimulation at the axon. Bottom, an average trace of EPSCs obtained from a PC–PC pair recording. Scale bars: 200 ms/50 pA. **g** Plot of the normalized peak amplitudes of aEPSCs at different frequencies and their single exponential fits. PC–PC EPSCs (20 Hz) are shown for comparison (dashed line). Data are represented as mean ± SEM. **h**, **i** Comparisons of paired-pulse ratio (interval: 50 ms; Wilcoxon rank-sum test) and release probability (two sample Student's $t$-test) of aEPSC and EPSC. *$P < 0.05$; **$P < 0.01$; ***$P < 0.001$; ns not significant

3.76 ± 0.17 to 2.74 ± 0.31 ms, $n = 7$ cells, $t_6 = 3.55$, $P = 0.01$, paired Student's $t$-test. Supplementary Figure 4). We also found that, in the experiments shown in Fig. 5, the decay time constant of baseline miniature EPSCs (mEPSCs) was significantly reduced after APV application (control: 9.6 ± 0.2 ms; APV: 5.1 ± 0.1 ms, $n = 8$ cells, $t_7 = 7.0$, $P = 0.0002$, paired Student's $t$-test; Fig. 5d, e). The decay time constant of aEPSCs, however, was not affected by APV (3.5 ± 0.2 vs. 3.6 ± 0.1 ms, $n = 8$ cells, $t_7 = -0.78$, $P = 0.46$, paired Student's $t$-test; Fig. 5d, e), further indicating the absence of NMDA component. The lack of NMDA receptors at autapses may prevent the occurrence of spike-timing dependent plasticity at these special synapses[30], despite the tight coupling and timing of pre- and postsynaptic APs at autapses.

**Autapses enhance burst firing.** What are the functional roles of PC autapses? Previous modeling[18] and experimental studies in *Aplysia*[20,21] have suggested a role of excitatory muscarinic autapses in the generation of persistent activity. In our slice preparation, however, we found no persistent activity that could outlast the evoked single or trains of APs in autaptic PCs, even when we depolarized the cells close to the AP threshold (from resting $V_m$ −65.3 ± 1.9 to −54.0 ± 2.5 mV, $n = 6$ cells; Supplementary Figure 5).

Other theoretical studies suggest that excitatory autapses may affect spiking activities[31–34]. To test this, we performed dynamic clamp experiments to examine changes in firing behavior after adding artificial autaptic conductances ($g_{aut}$) with a 1.4-ms delay (Fig. 6a). In axotomy experiments, the average aEPSC amplitude across cell population (−149 pA) and the maximum amplitude among individual trials (−524 pA) correspond to 2.1 and 7.5 nS with a driving force of 70 mV, respectively. Thus, we examined a range of $g_{aut}$ including conductances (2 and 4 nS) within the range of measured values, as well as those (8 and 10 nS) slightly

higher than the maximum value. In the presence of blockers for ionotropic glutamatergic and GABAergic synaptic transmission (Methods section), trains of APs with an initial instantaneous frequency ($f_{inst}$, $ISI^{-1}$) of 22.6 ± 1.2 Hz were evoked by depolarization steps. The delivery of $g_{aut}$ substantially increased the initial $f_{inst}$ to 37.6 ± 5.0 and 65.9 ± 12.5 Hz for 4 and 8 nS ($n = 14$ cells, $Z = 1.91$, $P = 0.0565$ for 4 nS; $Z = 2.78$, $P = 0.0054$ for 8 nS, Wilcoxon rank-sum test; Fig. 6b), respectively. Adding 4-nS ($n = 7$ cells) or 8-nS $g_{aut}$ ($n = 10$ cells) shortened the ISI at frequencies above 10 Hz, 8-nS $g_{aut}$ caused high-frequency AP doublets (>100 Hz) in ~50% of cells at an initial frequency of 50 Hz (Fig. 6b). These effects may result from a decrease in the size of after-hyperpolarization (AHP). In autaptic PCs, we compared the AP waveforms before and after the application of Kyn (1.5 mM). The half-width of APs showed no significant change (from 0.64 ± 0.04 to 0.63 ± 0.04 ms, $n = 8$ cells, $Z = 1.12$, $P = 0.26$, Wilcoxon signed-rank test). The post-spike AHP increased from 2.95 ± 0.39 to 4.88 ± 0.41 mV ($Z = -2.52$, $P = 0.011$, Wilcoxon signed-rank test). The CV of $V_m$ 5 ms after the AP peak decreased from 0.24 ± 0.07 to 0.11 ± 0.06 mV, but without significance ($Z = 0.84$, $P = 0.40$, Wilcoxon signed-rank test). Consistently, PCs with autaptic connections examined by laser axotomy or $SrCl_2$ treatment showed smaller AHP than those without autapses (peak amplitude: 3.02 ± 0.31 mV, $n = 76$ cells vs. 4.79 ± 0.39 mV, $n = 67$ cells, $Z = 3.60$, $P = 0.0003$, Wilcoxon rank-sum test; Fig. 6c, d), and autaptic PCs were more likely to generate high-frequency doublets (Fig. 6e).

To further investigate the role of autapses in regulating burst firing, we recorded layer-5 PCs with pipette solution containing 5 mM BAPTA (Fig. 6f) that could progressively block autaptic transmission (Fig. 2d). Previous studies showed that intracellular BAPTA would reduce the slow after-burst hyperpolarization but exert no effect on intrinsic bursting[35,36]. In our experiments, single electrical shocks evoked AP bursts on top of synaptic

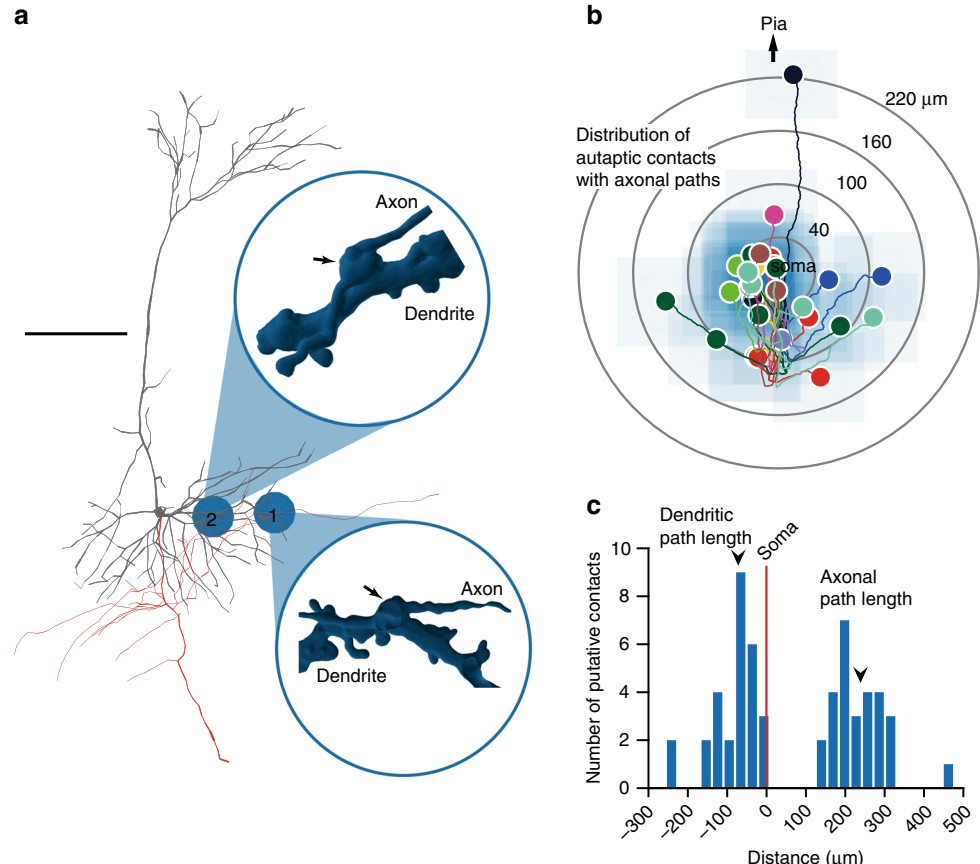

**Fig. 4** Autaptic contacts mainly locate at the basal dendrites. **a** Neurolucida three-dimensional reconstruction of a layer-5 PC. Gray, the soma and dendrites. Red, the axon and its collaterals. Inset, three-dimensional rendering of confocal images showing two putative autaptic contacts between axon and dendritic shaft. Scale bars: 100 μm. **b** Group data showing the locations of putative autaptic contacts with axonal paths relative to the soma ($n = 28$ contacts, 10 cells). **c** Distribution of dendritic and axonal path lengths of putative autaptic contacts

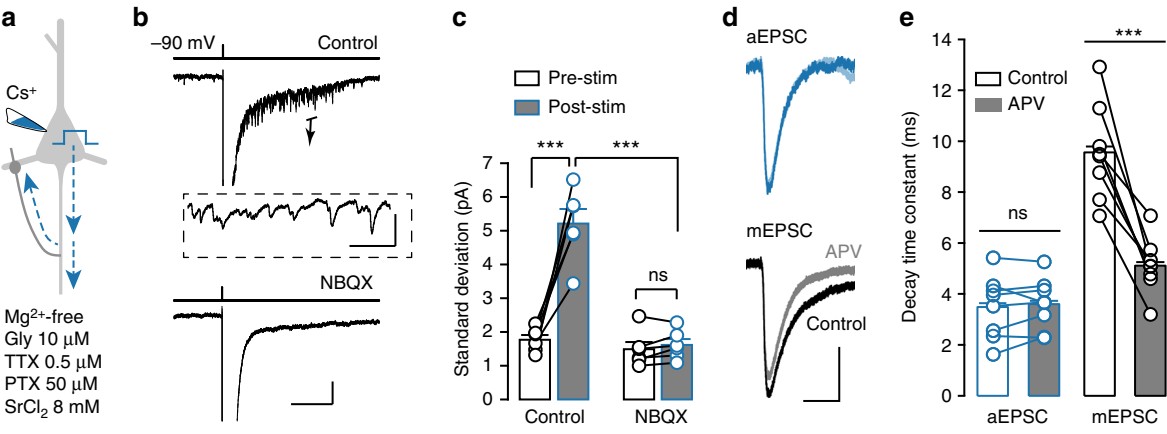

**Fig. 5** Autaptic responses contain no NMDA component. **a** Schematic drawing of layer-5 PC recording in $Mg^{2+}$-free ACSF (0 $Ca^{2+}$, 8 mM $Sr^{2+}$) but with a cocktail of drugs that enhances NMDA responses (Gly), blocks AP generation (TTX), and mIPSCs (PTX). The $Cs^+$ pipette solution makes the cell electrically more compact and allow efficient propagation of voltage pulses to autaptic contacts. **b** Desynchronized autaptic currents induced by somatic voltage pulses could be completely blocked by 10 μM NBQX. Scale bars: 1 s/50 pA. Inset: an expanded trace showing autaptic current events, scale bars: 50 ms/50 pA. **c** Group data showing current noise levels pre and post pulse stimulation, before and after NBQX application (paired Student's $t$-test). **d** Average traces of aEPSCs and baseline mEPSCs before and after APV (50 μM) application. Scale bars: 10 ms/10 pA. **e** Group data showing the effect of APV on the decay time constant of aEPSC and mEPSC (paired Student's $t$-test for both). Data are represented as mean ± SEM. ***$P < 0.001$; ns not significant

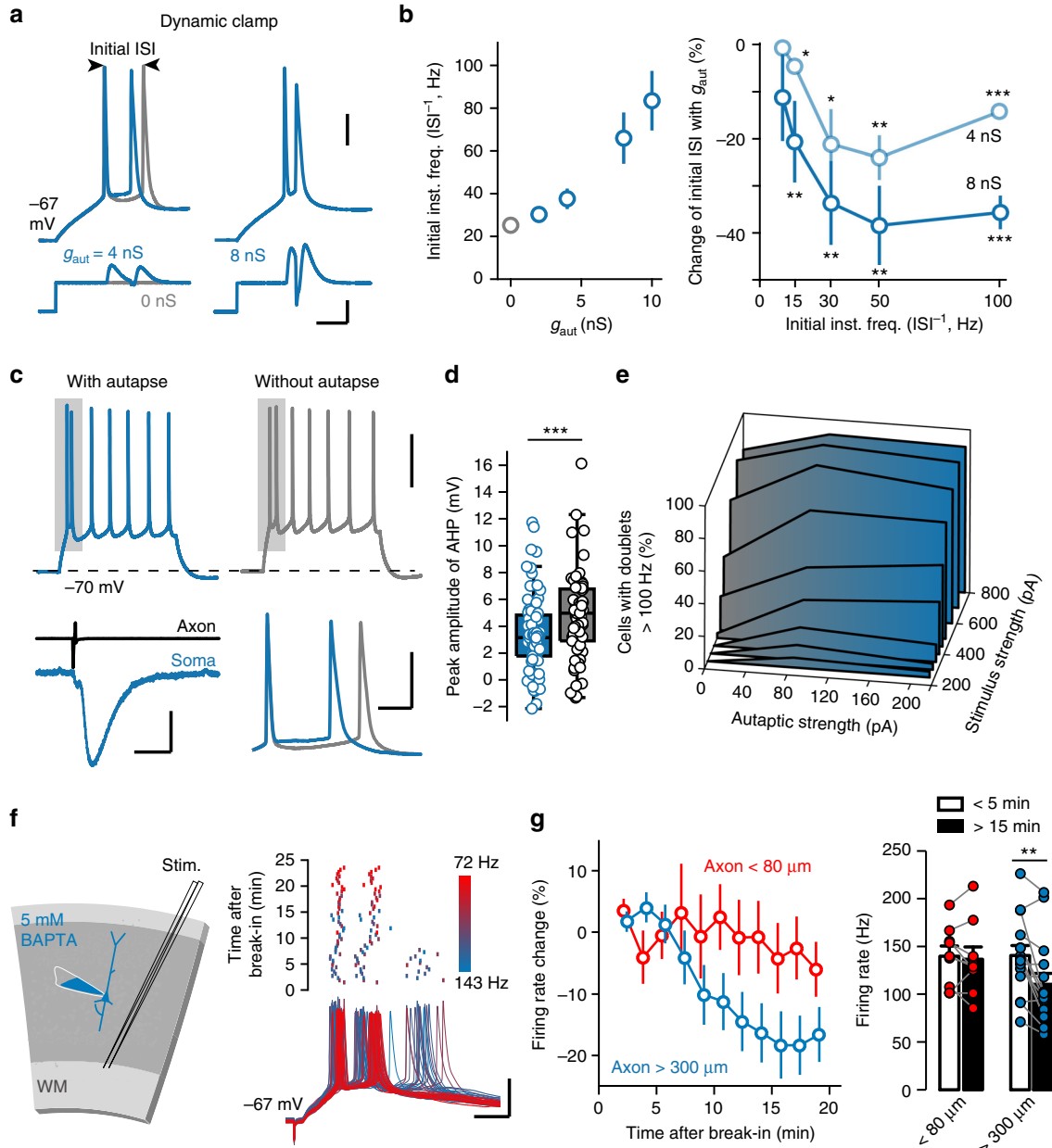

**Fig. 6** PC autapses enhance burst firing. **a** A representative dynamic clamp recording. Autaptic conductances ($g_{aut}$) were delivered to the recorded cell with a delay of 1.4 ms after each AP during step current injections. Note the shortening of the initial inter-spike interval (ISI) with the injection of 4-nS $g_{aut}$ and the burst firing with 8-nS $g_{aut}$. Scale bars: 20 ms/200 pA for current and 20 mV for $V_m$. **b** Left, plot of the initial instantaneous frequency as a function of $g_{aut}$. Right, percentage of changes in initial ISI with 4-nS or 8-nS $g_{aut}$ at different frequencies (Wilcoxon rank-sum test and paired Student's $t$-test). Data are represented as mean ± SEM. *$P < 0.05$; **$P < 0.01$; ***$P < 0.001$. **c** Top, two representative voltage responses of PCs with and without autapses to step current injections (200 pA, 500 ms) (scale bar: 40 mV). Bottom, an evoked aEPSC in the autaptic PC (left, scale bars: 10 ms/100 pA) and an overlay of the initial two APs from the gray boxes (right, scale bars: 10 ms/40 mV). **d** The average peak amplitude of AHP of PCs with and without autapses (Wilcoxon rank-sum test). Boxplots represent the median and interquartile range and whiskers represent 1.5× interquartile range. ***$P < 0.001$. **e** Plot of the percentage of cells with initial doublets higher than 100 Hz as a function of autaptic strength and stimulus strength. **f** Left, schematic of recording from a layer-5 PC with BAPTA-containing pipette solution while stimulating the slice at the border of layer VI and white matter (WM). Right, a representative recording from a PC (length of the axon trunk > 300 μm) showing a progressive reduction in intra-burst frequency with BAPTA infusion. Scale bars: 10 ms/ 40 mV. **g** Left, percentage of changes in firing rate in PCs with different axon-trunk lengths (<80 μm and >300 μm). Right, comparison of the firing rates within the initial 5 min and those 15 min later. Data are represented as mean ± SEM. **$P < 0.01$, paired Student's $t$-test

potentials (Fig. 6f). In PCs with axon trunks longer than 300 μm ($n = 15$ cells), the initial intra-burst frequency $f_{inst}$ was progressively reduced by 21.4 ± 4.8% ($t_{14} = 4.44$, $P = 0.0006$, paired Student's $t$-test; Fig. 6g) 15 min after break-in, i.e., establishing the whole-cell recording mode. As expected, PCs with short axons

(<80 μm, $n = 9$ cells) emitting no collaterals showed no significant reduction in intra-burst frequency (Fig. 6g).

**Autapses enhance PC responsiveness and coincidence detection.**
Next, we examined whether autapses alter neuronal

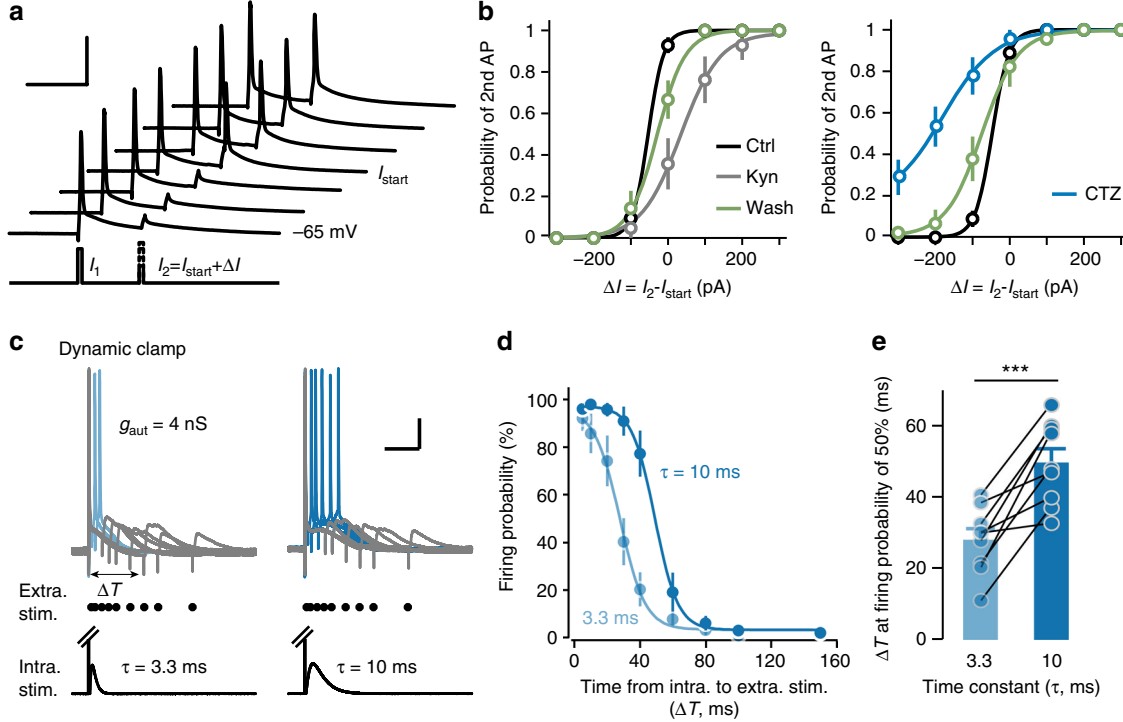

**Fig. 7** PC autapses promote coincidence detection. **a** Examining cell responsiveness to a second current pulse ($I_2$) shortly after the first stimulation ($I_1$) that evoked APs reliably. The inter-stimulus interval is 20 ms. $I_{start}$ is the current of the second pulse that evokes AP at a probability of ~0.9 in control condition. Changing $I_2$ from $I_{start}$ with increment of $\Delta I$ to examine the current threshold (i.e., AP probability at 0.5). Scale bars: 20 ms/50 mV. **b** Probability of the 2nd AP versus $\Delta I$ in different conditions (Kyn 1.5 mM and CTZ 100 μM). Data are presented as mean ± SEM. **c** Examine coincidence detection with AMPA-only ($\tau_{aut} = 3.3$ ms, left) or NMDA-containing autapses ($\tau_{aut} = 10$ ms, right) in short-axon PCs using dynamic clamp. The first AP was evoked by a brief current pulse (1 ms) and followed by $g_{aut}$ injection. Extracellular stimuli were delivered to the neighboring tissue after an interval of $\Delta T$. Scale bars: 50 ms/20 mV. **d** Plot of the firing probability of 2nd AP evoked by extracellular stimulation as a function of the $\Delta T$. **e** Group data showing the average $\Delta T$ at a firing probability of 50%. Data are represented as mean ± SEM. ***$P < 0.001$, paired Student's $t$-test

responsiveness to an immediate stimulation, similar to inhibitory autapses[11,37]. We applied paired current pulses (duration, 1 ms; interval, 20 ms) to the recorded PC every 4 s and allowed the first pulse ($I_1$) to evoke APs reliably (Fig. 7a). We varied the second pulse ($I_2$) starting from a current ($I_{start}$) that evoked APs with a probability of ~0.9 in control condition. In autaptic PCs, the application of Kyn (1.5 mM) significantly increased the threshold current (AP probability: 0.5) of $I_2$ by 92.7 ± 23.4 pA ($n = 14$ cells, $Z = -2.75$, $P = 0.006$, Wilcoxon signed-rank test; Fig. 7b), i.e., decreased the probability of the second AP. In contrast, enhancing aEPSCs by cyclothiazide (CTZ, 100 μM), a blocker of AMPA receptor desensitization, dramatically decreased the threshold current by 125.4 ± 21.5 pA ($n = 15$ cells, $Z = 3.23$, $P = 0.001$, Wilcoxon signed-rank test; Fig. 7b), i.e. increased the probability of the second AP. In PCs without autapses, both Kyn (change in threshold current: $-1.4 ± 12.6$ pA, $n = 22$ cells, $Z = 0.19$, $P = 0.85$) and CTZ ($-35.2 ± 24.6$ pA, $n = 12$ cells, $Z = 1.24$, $P = 0.21$, Wilcoxon signed-rank test) showed no significant effect on the threshold current. These results suggest that autaptic responses increase neuronal responsiveness to a subsequent stimulus and support the modulation of burst firing by autapses (Fig. 6).

The time window during which increased responsiveness occurs is most likely limited by the kinetics of the autaptic currents, and the lack of NMDA receptors at autapses (Fig. 5) may result in PCs as strict coincidence detectors with a narrow time window. To test this, we also employed dynamic clamp to apply AMPA-only ($\tau_{aut} = 3.3$ ms) and NMDA-containing autaptic conductances, $g_{aut}$ ($\tau_{aut} = 10$ ms), after AP generation evoked by brief current injections (Fig. 7c). To avoid the interference of intact autapses, we recorded from short-axon PCs. With the slower NMDA-$g_{aut}$, extracellular stimulation generated a second AP during a much wider time window than with the fast AMPA-$g_{aut}$. The probability of generating a second AP by extracellular synaptic stimulation in the presence of AMPA-only $g_{aut}$ was dramatically decreased with longer inter-stimulus interval $\Delta T$ (Fig. 7d). The curve of firing probability versus $\Delta T$ showed a left shift by 21.6 ± 3.9 ms ($n = 9$ cells, $t_8 = -5.49$, $P = 0.0006$, paired Student's $t$-test; Fig. 7e), indicating that AMPA-only autapses facilitate coincidence detection of incoming synaptic inputs strictly time-locked to AP firing of the PC itself.

## Discussion

Here we provide physiological evidence in acute brain slices showing the existence of functional autapses in neocortical PCs. Importantly, activation of these autapses causes large postsynaptic responses, which promote neuronal responsiveness and burst firing. Due to the lack of NMDA component, AMPA-only autaptic response enables strict coincidence detection of self-activity and incoming synaptic inputs in PCs. Our results suggest that the autaptic responses could survive the previously proposed shunting effect of APs[38] and exert net excitation rather than inhibitory shunting effect on subsequent AP generation[11,37]. Our findings show that functional glutamatergic autapses are abundant in mouse and human neocortical circuits.

Because neurons in culture bear enormous number of autapses, much more than those found in the intact brain, autaptic contacts in vivo often are considered as a wiring error[37]. However, the sparseness of autapses does not apply to all types of cells. Cortical GABAergic interneurons, for example, possess massive autaptic connections[8], activation of these autapses produces GABA_A

receptor-mediated postsynaptic responses and regulates action potential generation[11] and spike-timing precision[10], indicating autaptic connections are not only functional but also involved in neuronal signal processing. In these studies, because of the short half-width APs in GABAergic cells, autaptic responses do not show much overlap with action currents and are readily detected using high-$Cl^-$ pipette solution in voltage-clamp mode.

Although early anatomical studies found autapses in excitatory PCs in the neocortex[1,6,8], recording autaptic responses in slices seemed impossible because they could be masked by broad, long half-width APs of PCs[6]. In our simultaneous soma and axon recording from a single cell, axotomy blocked AP invasion from the axon to the somatodendritic compartments and unmasked autaptic responses. Their brief onset latency shows that these are monosynaptic responses. In addition, desynchronizing synaptic transmission with $Sr^{2+}$ also revealed the occurrence of autaptic currents. In the presence of TTX to block AP generation also by other neurons in the network, depolarizing pulses in the soma could still evoke autaptic responses, further confirming monosynaptic transmission. To the best of our knowledge, these results are the first piece of physiological evidence showing PC autapses are functional in the neocortex.

Previous morphological studies obtained inconsistent results on the occurrence frequency of autapses in cortical PCs. van der Loos and Glaser[1] found autaptic connections in 50% of the traced Golgi-impregnated PCs in rabbit occipital cortex, the percentage of PCs with autapses were underestimated because only half of the PC dendrites were preserved in their 100-μm-thick sections. Anatomic examination of biocytin-labeled cells in thick cortical slices (300–400 μm) revealed a large proportion of layer-5 PCs (~80%) in the developing rat neocortex form autaptic contacts[6], but very few autaptic contacts were found in layer 2–4 PCs of the adult cat neocortex[8]. Differences in species, age, cortical region, and layer may lead to the inconsistent observations. Considering that autapses tend to form in specific types of cells, for example, they selectively occur in fast-spiking but not in low-threshold spiking inhibitory interneurons[8,11], we speculated that the inconsistency of autapse abundance in PCs could be attributable to differences in cortical layers. Indeed, we obtained autaptic currents in a large proportion of PCs in layer 5, fivefold more than those in layer 2–3 (Fig. 2). Even among layer-5 PCs, those projecting to subcortical regions but not cPFC-projecting PCs tend to form autapses. What determines the selectivity of autapse formation remains to be further examined.

Prolonged self-excitation may be critical for the generation of persistent activity. Modeling studies have proposed that autaptic self-excitation could cause neuronal persistent activity. Experiments in *Aplysia* indeed demonstrated that cholinergic autapses in motor neuron B31/32 produce long-lasting depolarization plateau and persistent activity via the activation of muscarinic receptors. In our experiments, however, either single AP or a burst of APs could not trigger the generation of persistent activity, even when we depolarized the cell to just below the firing threshold. The brief synaptic delay (<2 ms) and short-duration AMPA-only responses could be attributable to the failure of generating persistent activity in PCs.

The selective formation of autaptic connections suggests that they may play important roles in specific groups of cells. In fast-spiking cells that discharge precisely during network oscillations such as theta and gamma rhythms, inhibitory autapses reduce the jitter of APs and thus increase the spiking precision[10]. The inhibitory effect on AP generation is exerted through autaptic shunting inhibition mediated by an increase in membrane conductance[11,39,40]. Similar inhibitory function for excitatory autapses has been speculated in early studies[1]. Our results, however, clearly show that activation of PC autapses facilitates the

generation of the following AP and increases neuronal responsiveness to incoming inputs, indicating a role of self-excitation. Dynamic clamp results indicate a role of autapses in promoting burst firing in layer-5 PCs. Because of short-term depression, we speculate that autapses may exert stronger effects on neuronal signaling at low frequencies and at the onset of higher frequencies.

A well-known mechanism for burst firing in layer-5 PCs is the activation of dendritic voltage-gated $Ca^{2+}$ channels upon the arrival of backpropagating APs[41]. Voltage-gated $Na^+$ channels promote AP backpropagation to the apical dendrites and allow sufficient depolarization to reach the threshold of dendritic $Ca^{2+}$ spikes, which would facilitate the generation of AP bursts. Previous studies also revealed an important role of persistent $Na^+$ current in burst firing[42,43]. Blocking (or enhancing) $Na^+$ sodium current could abolish (or promote) burst firing in layer-5 intrinsic bursting PCs. A recent study has indicated that the activation of persistent $Na^+$ current at the first node of Ranvier may contribute substantially to the generation of AP bursts[44]. High-frequency bursts would not occur if the axon was cut before the first node or $Na^+$ channels at the first node were blocked. Since these experimental manipulations block AP conduction along the axon and prevent autaptic responses, the results also support a role of PC autapses in AP bursts. However, it remains to be further examined whether autapses also contribute to AP bursts in other brain regions, including the somatosensory cortex.

Burst firing in PCs may turn unreliable synaptic transmission to be reliable and cause supralinear summation of postsynaptic potentials[45,46], leading to increases in functional connectivity and network synchronization between remote cortical regions[47–49]. Self-excitation autapses could thus serve as a potential mechanism for burst firing in PCs, facilitating the formation of functional networks such as attention-associated networks[49]. Autapses may also serve as a key target of neuromodulators[50,51], such as dopamine, norepinephrine and acetylcholine, and play important roles in regulating cortical functions.

In summary, our results indicate that PC autapses are not a wiring error and prodigal structures in the neocortex, rather a functionally important circuit element of the brain[37,50]. Thus, this fundamental structure should be considered in dissecting the mechanisms of cortical functions and constructing computational models of the cerebral cortex[52].

## Methods

**Slice preparation.** For each experiment, animals of either sex with similar ages were assigned randomly. The investigators were not blinded to experimental conditions and outcome assessment. The use and care of laboratory animals complied with the guidelines of the Animal Advisory Committee at the State Key Laboratory of Cognitive Neuroscience and Learning, Beijing Normal University. The Animal Advisory Committee also approved the study protocol. C57/B6 mice (postnatal day 13–21) were killed by rapid decapitation and brain tissues were immediately dissected out. Adult mice (8 weeks old) were also used in experiments examining the abundance of autapses. In ice-cold sucrose-based slicing solution (normal ACSF listed below but with NaCl replaced with equiosmolar sucrose) that had been bubbled with 95% $O_2$ and 5% $CO_2$, tissue blocks from mice were cut coronally with a vibratome (Leica VT1000S). Slices (300-μm thick) were collected and incubated at 35 °C in aerated artificial cerebrospinal fluid (ACSF) containing (in mM): NaCl 126, KCl 2.5, $MgSO_4$ 2, $CaCl_2$ 2, $NaHCO_3$ 26, $NaH_2PO_4$ 1.25, and dextrose 25 (315 mOsm, pH 7.4). After 90-min incubation, slices were then incubated at room temperature until use.

The use of human brain tissue and the study protocol were approved by the Ethics Committee at Sanbo Brain Hospital, Capital Medical University. We have complied with all relevant ethical regulations relating to the use of resected human brain tissue in research. Clinical investigations were conducted according to the declaration of Helsinki. The patients and their relatives provided written informed consent before surgeries. Cortical tissues were obtained from two patients (Asian, male, 31 and 39 years old) with intractable epilepsy, removal of these tissues was essential for the treatment of epileptic seizures. The 31-year-old patient had a 10-year history of complex partial seizures. Electrographic seizures were captured by the intracranially implanted electrodes, and multiple locations responsible for

epileptogenesis were identified, including the right frontal-lobe cortical regions. This patient was diagnosed as tuberous sclerosis by post-operative pathological examination. The 39-year-old patient had suffered from epileptic seizures for 30 years, and in the recent months before surgery he experienced 4–5 epileptic events per week. Magnetic resonance imaging (MRI) and EEG monitoring suggested a right frontal region responsible for epileptogenesis. This patient was diagnosed as focal cortical dysplasia (FCD) by post-operative pathological examination. Immediately after surgical resection, cortical tissues were immediately immersed into the 0 °C aerated sucrose-based slicing solution (see above) and transported to the laboratory for slice preparation (same procedures as for mouse tissue). The transportation took approximately one hour. The cutting orientation of human tissues was unknown, but the blade was adjusted perpendicular to the pia of cortical blocks to minimize damage of the vertically oriented PCs.

**Electrophysiological recording**. Slices were transferred to the recording chamber and perfused with aerated ACSF (34-35 °C) at a rate of 1.2 ml/min; drugs were applied through bath perfusion. Cortical neurons were visualized under upright infrared differential interference contrast microscope (BX51WI or BX61WI, Olympus). Unless otherwise stated, we chose to perform recordings from large layer-5 PCs, which were identified by their pyramid-shaped soma, a single thick apical dendrite and an axon projecting toward the white matter. The impedances of patch pipettes for somatic and axonal recordings were 3–5 MΩ and 6–10 MΩ, respectively, when filled with the internal solution (in mM): K-gluconate 145, MgCl₂ 2, Na₂ATP 2, HEPES 10, EGTA 0.2 (286 mOsm, pH 7.2). Biocytin (0.2%) was also added for post hoc avidin staining. To obtain dual whole-cell recording from the soma and the axonal bleb of PCs, we formed recording from the soma first with a pipette filled with an internal solution containing Alexa Fluo-488 (100 μM, green), the axonal structure and the terminal bleb formed during slicing procedures could be visualized 3–5 min later, we could then obtain recording from the bleb using another patch pipette containing Alexa Fluo-594 (50 μM, red). Both recordings were in voltage-clamp mode and the holding potential was −70 mV. Individual action currents (or APs) were elicited by voltage pulses (100 mV, 0.7–1.0 ms, 1–2 Hz) at the axonal bleb, changes in current traces were monitored at both recording sites. In some experiments, axons were recorded in current-clamp mode and stimulated by current pulses.

As soon as dual soma-axon recording was achieved, two-photon laser axotomy was carried out at a location just beyond the axon initial segment (60–80 μm away from the soma) to disconnect the axon and the somatodendritic compartments. Thus backpropagating APs could not reach the somatodendritic compartments and therefore aEPSCs arrived at the dendrites could be unmasked. Imaging of recorded cells and axotomy was achieved under a two-photon laser scanning microscope equipped with a water immersion objective (×40, NA 0.8) and a mode-locked Ti:Sapphire laser (Mai Tai DeepSee, Spectra-Physics) set at 840 nm (repetition rate: 80 MHz; pulse width: 80 fs). We increased the laser intensity and applied brief bleaching pulses (50–200 ms in duration) to the selected axon location until axotomy was achieved, as indicated by a sudden change of the holding current and absence of somatic action currents. Because the dye diffusion was blocked after axotomy, the somatodendritic and the axonal compartments were labeled by two distinct dyes respectively. Imaging data were acquired with Fluoview FV 1200 (Olympus) and further analyzed by ImageJ and MATLAB (MathWorks, USA).

In the Sr²⁺ experiments examining autaptic connections, we added 8 mM SrCl₂ to the ACSF but reduced the concentration of CaCl₂ and MgSO₄ to 1 mM. Trains of 4 voltage pulses (1 ms in duration, 100 mV, 20 Hz) every 20 s were applied to the cell through the somatic recording pipette (filled with the K⁺-based internal solution) to evoke action currents. The presence of desynchronized synaptic currents during and after the train stimulation indicates the existence of autapses. Because axons were cut before branching, PCs with short axons (<80 μm) in these experiments were not included for data analysis.

In another sets of Sr²⁺ experiments examining whether autapses express NMDA receptors, we recorded asynchronous aEPSCs in Mg²⁺-free ACSF but in the presence of 10 μM glycine or 100 μM D-serine. We added 8 mM SrCl₂ to the bath but omitted CaCl₂. Tetrodotoxin (TTX, 0.5 μM) was used to minimize synaptic inputs from other cells and thus pharmacologically isolate the recorded PC from the network. Picrotoxin (PTX, 50 μM) was added to block GABAₐ receptor-mediated miniature inhibitory postsynaptic currents (mIPSCs). A Cs⁺-based pipette solution (in mM: CsMeSO₃ 138, CsCl 3, MgCl₂ 2, Na₂ATP 2, HEPES 10, EGTA 0.2; 285 mOsm, pH 7.2) was used to block K⁺ conductances, making the cell electrically more compact, and allowing effective propagation of voltage pulses to presynaptic terminals to evoke autaptic responses. At a holding potential of −70 or −90 mV, we applied voltage pulses (100–200 mV, 10–1000 ms) every 25 s to the soma and monitor the occurrence of asynchronous aEPSCs. Properties of the evoked autaptic events were then compared with those in the presence of 50 μM APV, an NMDA receptor antagonist. Individual non-overlapping aEPSC events and mEPSCs (occurring at baseline before voltage pulses) were included for analysis. To obtain desynchronized EPSCs from recurrent synapses, we delivered single electrical shocks (0.1 ms in duration, 10–20 μA) to the neighboring tissue of the recorded layer-5 PCs under similar condition described above but with reduced concentration of Sr²⁺ (3 mM) and no TTX. These experiments were performed at room temperature.

Paired-pulse stimulation (1 ms in duration for each pulse, interval: 20 ms) was employed to examine the responsiveness of PCs with and without autapses. The

first pulse was adjusted to evoke APs reliably. To examine the input–output curve of the second pulse, we adjusted the current amplitude to a start level ($I_{start}$) that evoked APs at a probability of ~0.9 in control condition and then generated the input–output curve with decrement or increment of 100 pA. The mid-point current that evokes AP with a probability of 0.5 is considered as the threshold current. At the end of each recording, we switched the bath to Sr²⁺-containing ACSF to examine whether the recorded PC possessed autapses. Note that the blockade of autaptic responses by Kyn is reversible.

Simultaneous whole-cell recordings were obtained from two neighboring PCs with a distance less than 70 μm to examine unitary properties of unitary EPSCs in recurrent synapses. Brief voltage pulses (0.7–1.0 ms, 100 mV) were used to evoke single action currents in the presynaptic PC and induce EPSCs in the postsynaptic PC. The short-term synaptic plasticity was also examined with trains of pulse stimulation every 10 s. Properties of PC–PC EPSCs were compared with those of aEPSCs obtained in axotomy experiments. The onset latency was measured as the time between the peak of presynaptic action current and the onset of EPSCs, and the onset was determined by the crossing of the baseline current and the linear fit of EPSC rise phase. The rise time was measured from 20 to 80% of peak, and the decay time constant was obtained by single exponential fit. To remove the preceding artifacts (i.e., residual backpropagating action currents), we fitted individual aEPSCs with an alpha-synapse function and then measured the rise time and onset latency. To obtain the decay time constant of aEPSC, we fitted the raw traces with a single exponential function.

To examine the effect of autapses on burst firing induced by synaptic stimulation, we placed a bipolar electrode at the border of layer 6 and the white matter and delivered single electrical shocks (0.1 ms in duration) to the tissue every 25 s. The stimulation intensity was adjusted to allow burst firing at a frequency of 100–150 Hz. The bath solution contained PTX and APV for blocking GABAₐ and NMDA receptors, allowing synaptic transmission via AMPA/kainite receptors. The K⁺-based pipette solution contained 5 mM BAPTA. The firing frequency was monitored since the formation of whole-cell recording (i.e., membrane break-in).

Dynamic clamp was achieved using CED power 1401 and Signal software. We performed whole-cell recording from the PC soma and adjusted the injected 500-ms step currents to evoke APs at a mean frequency of 14 Hz with an initial instantaneous frequency of ~20 Hz. We then monitored $V_m$ changes and AP generation before and after the insertion of autaptic alpha-synapse conductances ($g_{aut}$ amplitude: 0–10 nS; $\tau_{aut}$: 3.3 ms; reversal potential: 0 mV). These artificial conductances were applied after each AP with an onset latency of 1.4 ms (from the time when the AP surpasses 0 mV to the onset of $g_{aut}$). We measured the initial ISI and its changes after the insertion of autaptic conductance. To avoid the interference from autapses and synapses, we applied CNQX (20 μM), APV (50 μM), and PTX (50 μM) to block the fast synaptic transmission. In the experiments that examined the coincidence detection of synaptic inputs in PCs, the bath contained APV and PTX but no CNQX. To avoid the effect of intact autapses, we recorded PCs with short axons ( < 80 μm) showing no collaterals. The first AP was induced by current injection (1 ms in duration, 2–3 nA) and followed by $g_{aut}$ injection. We adjusted the kinetics of $g_{aut}$ to mimic AMPA-only ($\tau_{aut}$ = 3.3 ms) and NMDA-containing autapses ($\tau_{aut}$ = 10 ms). After an interval of $\Delta T$ from intracellular stimulation, a single electrical shock (0.1 ms) was delivered by a glass electrode placing in the neighboring tissue (50–80 μm from the soma of the recorded cell) to evoke synaptic inputs. We varied $\Delta T$ from 5 to 150 ms and monitored the probability of the evoked second AP.

Voltage and current traces were low-pass filtered at 10 kHz and sampled at 20 kHz using Spike2 or Signal software (Cambridge Electronic Design). The liquid junction potential was not corrected for $V_m$ values showing in figures and the main text.

**Cell staining and reconstruction**. Slices were fixed with 4% paraformaldehyde (more than 12 h) and stained with Alexa Fluor-488-conjugated avidin. The z stack images (0.7 μm between successive images) of individual cells were acquired by a confocal microscope (A1 plus, Nikon) equipped with a ×60 objective. Three-dimension reconstruction of the cells was performed using a Neurolucida system (MBF Bioscience) to form a continuous 3D representation of the entire cell structure. Individual sites where the distance between an axon and a dendrite was less than 1.4 μm were visually inspected and then imaged with higher magnification (×100 objective, z spacing of 0.2 μm, a scan zoom factor of ×5). Geometrical contacts are those without an obvious decrease in the fluorescence intensity between the axon and the dendrite. Boutons are axonal sites showing significant swelling of at least threefold the average axonal width. Putative autaptic contact is a geometrical contact between a bouton and a dendritic spine or shaft. Identification of putative autaptic contacts was performed without knowledge of functional autaptic connectivity.

**Statistical analysis**. Data analysis was performed using MATLAB and Spike2 software. All measurements were taken from distinct samples. No statistical methods were used to predetermine sample size. We determined the sample size based on numbers previously reported. To compare between groups and conditions, we obtained data from at least five cells from different animals. No samples were excluded from the analysis. Unless otherwise stated, data are presented as mean ± SEM, error bars in figures also represent SEM. For paired observations,

two-tailed paired Student's *t*-test was used to examine the significance level of difference between groups with normal distribution ($P > 0.05$, Shapiro–Wilk's test); non-normal data were compared by Wilcoxon matched-pairs signed-rank test. For two independent observations, two samples of unpaired Student's *t*-test (two-tailed) were used for data that passed the normality test; non-normal data were compared using Wilcoxon rank-sum test. Differences were considered to be significant if $P < 0.05$.

## Data availability

Data is available from the corresponding author upon request.

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

## Acknowledgements
We thank John R Huguenard and Huibert D Mansvelder for helpful suggestions. This work was supported by the National Natural Science Foundation of China Projects (31430038, 31630029, 31661143037 and 81571275).

## Author Contributions
Y.S. initiated the project and designed the experiments. L.Y., R.Z., W.K. Q.H. and Y.S. performed electrophysiological recordings, cell reconstruction and data analysis; Y.Z., J. L., B.W. and Z.M. contributed to path-clamp experiments. Y.-s.L. and M.J.R. contributed to data analysis. G.L. and T.L. contributed to human tissue processing. Y.S., W.K., Q.H. and L.Y. wrote the manuscript.

## Additional information

**Competing interests:** The authors declare no competing interests.

