## [Peer Review File · Nature Communications]

Summary of the revision (To all reviewers):

In response to comments of the reviewers, we have performed the following new experiments and analyses, and added more discussion:

1. On the effect of axotomy on autaptic response waveform. We compared the waveform of spontaneous EPSCs before and after the axotomy and found no significant difference in the peak amplitude and kinetics, suggesting that the waveform of aEPSCs would also not change after axotomy.
2. On the contribution of NMDA receptors to aEPSC and PC-PC EPSC. We performed new experiments using a more specific antagonist for non-NMDA receptors, NBQX, and found it could completely block autaptic events. In addition, we employed extracellular stimulation to evoke desynchronized EPSCs from recurrent synapses and found that APV could significantly reduce the decay time constant.
3. On the number of putative synaptic and autaptic contacts. We performed 3-D reconstruction of connected PC-PC pairs and found that the putative synaptic contacts are approximately 3 times less than autaptic contacts.
4. On the quanta size and kinetics of aEPSCs and PC-PC EPSCs. We have now provided the amplitude, the rise time and decay time constant.
5. On the mechanism underlying burst firing. We have added more discussion on the role of dendritic voltage-gated calcium and sodium channels in promoting burst firing. The experimental manipulations in Kole's study (Neuron, 2011) also support a role of autapse in AP burst.

Responses to Reviewer #1 (Reviewers' comments in italic):

This very elegant manuscript directly demonstrates the presence and function role of excitatory autaptic contacts predominately in subcortical projecting pyramidal neurons. The use of dual somato-axonal bleb recording and targeted axon cutting very nicely demonstrates the presence of autaptic contacts in layer 5 pyramidal neurons. The use of Sr to desynchronize transmitter release provides additional support and an opportunity to survey a large number of neuronal classes in rodent and human. I am very persuaded by the author's data, and strongly support publication. Yet I am puzzled with the contrast between this work and Kole 2011 (Neuron), where using similar

techniques it was argued that the first node plays a role in driving burst firing, similar to the present study, but reports no evidence of autaptic input. The work of Kole is in some ways consistent with the present work i) axotomy before the first node, prevents burst firing, and ii) block of sodium channels in this area blocks burst firing. The authors have not referenced this paper, and at the very least in revision should dedicate detailed discussion to this issue.

We thank the reviewer for the comments on the methods and findings of this study. According to the reviewer's suggestion, we have now added more discussion to the manuscript about the mechanisms underlying the burst firing in neocortical pyramidal cells. A well-known mechanism is that the burst firing results from the activation of dendritic voltage-gated calcium channels upon the arrival of backpropagating action potentials (APs) (Williams and Stuart, 1999). Voltage-gated sodium channels promote AP backpropagation to the apical dendrites and allow sufficient depolarization of the dendrites to reach the threshold of calcium spikes, which will facilitate burst firing. Previous studies also revealed an important role of persistent sodium current in burst firing (Franceschetti et al., 1995; Mantegazza et al., 1998). Blocking (or enhancing) persistent sodium current could abolish (or promote) burst firing in layer-5 intrinsic bursting PCs.

A recent study (Kole, 2011) has indicated that the activation of persistent sodium current at the first node of Ranvier (~100 μm away from the axon initial segment) may contribute substantially to the generation of high-frequency AP bursts. The burst firing in layer-5 PCs in neocortical slices would not occur if the axon was cut (axotomy) before the first node or sodium channels at the first node were blocked (Kole, 2011). In Kole's study, the contribution of autapses could not be excluded since both experimental manipulations blocked AP conduction along the axon to reach the autaptic contacts.

[Redacted]

Responses to Reviewer #2:

The manuscript by Yin et al. investigates the existence and subsequent role of autapses in neocortical pyramidal neurons. Using dual patch clamp electrophysiology, the authors found autapses in both rodent and human pyramidal neurons, which they showed enhanced burst firing, neuronal responsiveness and coincidence detection of synaptic inputs. This quality of the recordings are high, and the manuscript is generally well written. However, the impact of the findings is questionable, especially considering excitatory autapses have been previously reported in hippocampal pyramidal neurons (albeit in cultures).

We thank the reviewer for the evaluation of our recordings and the manuscript writing. We believe that our findings are novel and will enrich our understanding on the role of glutamatergic autapses in cortical functions. Indeed, previous studies reported that excitatory autapses are abundant in hippocampal pyramidal cells in culture systems.

[Redacted]

Since autapses could be hardly found in intact cortical tissue in previous studies, they are considered as wiring errors or aberrant and redundant structures if they do exist in the brain. In this study, we found that excitatory autapses selectively form in a subpopulation of neocortical PCs and play important roles in neuronal signaling.

The 'individual' aEPSC amplitudes are 5-10pA after addition of Sr²⁺ in the bath (Figure 2). However, the average aEPSC amplitude is reported as ~ 149 pA (Figure 4). Therefore, presuming linear summation of the EPSCs at the soma, that equates to at least 15 autapses. This is fivefold more than predicted via fluorescence imaging. Of course, this is hard to assess as the desynchronized amplitudes are not included in the text, but can the authors explain the discrepancy. Following from this, the manuscript is surprisingly lacking in detail. What is the average half width? Etc...

Since a particular synapse may have many release sites, we could not estimate the number of autaptic contacts from the aEPSC amplitude. We have now provided the average peak amplitude and half-width of the desynchronized aEPSCs in Sr²⁺-ACSF. These desynchronized autaptic events had an average peak amplitude of -19.3 ± 2.6 pA and an average half-width of 4.31 ± 0.38 ms (rise time: 0.91 ± 0.03 ms; decay time: 3.92 ± 0.40 ms). We also provided these values of human PC aEPSCs (average peak amplitude: -15.3 ± 1.3 pA; average half-width: 5.62 ± 0.15 ms; rise time: 1.09 ± 0.02 ms; decay time: 6.35 ± 0.32 ms). Please also note that

the average peak amplitude was relatively larger than that (i.e. quanta size) in the experiment with TTX (Figure 5). In the presence of TTX, the average peak amplitude of desynchronized PC-PC EPSCs was -7.96 ± 2.68 pA ($n = 5$), similar to those of aEPSCs (-8.49 ± 2.56 pA, $t_8 = 0.14$, $n = 5$, $P = 0.89$, two sample Student's t -test).

The spatial resolution of the imaging is not adequate to conclusively identify functional autaptic contacts. Simply because an axon is in close proximity, does not infer functional connectivity. To address this, EM or Calcium imaging would need to be performed.

We thank the reviewer for pointing out this. Indeed, an axon in close proximity to a dendrite (i.e. geometrical contacts) does not infer functional connectivity. In this study, we identified the putative autaptic contacts in cells stained with Alexa Fluo-488-conjugated avidin using procedures reported previously (Kalisman et al., 2005). In 3-D reconstruction analysis (see Methods), we first identified the geometrical contacts, which are those without an obvious decrease in the fluorescence intensity between the axon and the dendrite. Only those geometrical contacts with axon boutons (i.e. axonal sites showing significant swelling of at least 3-fold the average axonal width) were considered as putative autaptic contacts. Identification of putative autaptic contacts was performed without knowledge of functional autaptic connectivity.

[Redacted]

Since this is a slice preparation, axons are severed (as clearly seen in figure 1a and 3a). Is there a correlation with the length of axon and the number of putative autapses? Due to the increased likelihood of severing the axon and/or dendrite as you venture further from the soma, it is quite dangerous to do a direct comparison between the location of putative autapses on basal and apical dendrites.

Indeed, PC axons are cut at some point during slicing procedures. We performed analysis as suggested by the reviewer (see the figure showing below). We only found a weak correlation ($R = 0.28$, $P = 0.06$) between the number of putative autaptic contacts and the total axonal path length.

[Redacted]

We have now changed the statement “putative autapses favored basal dendrites...over apical dendrites” to “most of the putative autapses formed at the basal dendrites ($n = 24/28$ contacts)”

[Redacted]

As shown by Kole 2011, axotomy and severed axons itself alters high-frequency burst generation. Can the authors explain how this alters their interpretation of their results and a discussion should be included in the manuscript.

According to the suggestion from this reviewer and reviewer #1, we have now added more discussion to the manuscript about the mechanisms underlying the burst firing in neocortical pyramidal cells. A well-known mechanism is that the burst firing results from the activation of dendritic voltage-gated calcium channels upon the arrival of backpropagating action potentials (APs) (Williams and Stuart, 1999). Voltage-gated sodium channels promote AP backpropagation to the apical dendrites and allow sufficient depolarization of the dendrites to reach the threshold of calcium spikes, which will facilitate burst firing. Previous

studies also revealed an important role of persistent sodium current in burst firing (Franceschetti et al., 1995; Mantegazza et al., 1998). Blocking (or enhancing) persistent sodium current could abolish (or promote) burst firing in layer-5 intrinsic bursting PCs.

A recent study (Kole, 2011) suggests that the activation of persistent sodium current at the first node of Ranvier (~100 μm away from the axon initial segment) may contribute substantially to the generation of high-frequency AP bursts. The burst firing in layer-5 PCs in neocortical slices would not occur if the axon was cut (axotomy) before the first node or sodium channels at the first node were blocked (Kole, 2011). In Kole's study, the contribution of autapses could not be excluded since both experimental manipulations blocked AP conduction along the axon to reach the autaptic contacts.

[Redacted]

Due to the uncontrolled nature of the removal of the human tissue, it is often difficult to obtain preparations with neurons in plane. Therefore, the PC morphology would be more variable than in the mouse where cutting angle can be pre-determined to slice in plane. This may influence the percentage of cells which appear to have autapses, and would explain the decreased percentage. Following this, the authors should include reconstructions of their human recordings.

[Redacted]

The aEPSC amplitude is a summation of the activation of multiple synapses. This is evident not only in the morphological reconstructions, but also the varying amplitudes of the aEPSCs. Do the authors know how many putative connections the PC-PC make? To be able to compare EPSC amplitudes, the authors should do Sr experiments in PC-PC pairs and not simply state that aEPSCs are comparatively giant.

According to the reviewer's suggestion, we have performed more recordings and morphological analysis from synaptically connected PC-PC pairs. We found that the average number of putative synaptic contacts between PCs is 1.14 ± 0.14 ($n = 7$ pairs), less than those of autaptic contacts per cell (2.80 ± 0.59). Our Sr^{2+} experiments (in the presence of TTX) revealed that the average peak amplitude of desynchronized PC-PC EPSCs was -7.96 ± 2.68 pA ($n = 5$), similar to those of aEPSCs (-8.49 ± 2.56 pA, $t_8 = 0.14$, $n = 5$, $P = 0.89$, two sample Student's t -test). These results suggest that the relatively large aEPSCs could be due to more autaptic contacts and higher release probability (Fig. 4i). We have now added these results to the revised manuscript.

More details are required in the introduction. It is confusing to talk about layers without talking about regions as different cortical regions have layers which receive information from different sources. Perhaps this explains the inconsistent observations stated in the introduction. Further, the data should be presented in a different manner where the kinetics and characteristics of the autaptic response is presented before their synaptic location.

We have now modified the introduction and clarified the cortical regions, age and species in those cited early studies. We agree with the reviewer that cortical regions may also explain the inconsistent observations on the abundance of autapse in neocortical PCs. The abundance of PC autapses in different cortical regions (even other brain structures), species and age remains to be further examined.

In accordance to the reviewer's suggestion, we have now presented the kinetics and characteristics of autaptic responses before the section of the autapse location.

The use of the terms 'corticocortical' and 'subcortically-projecting' PCs is misleading. Of course, the prefrontal cortex projects to other cortical regions than the ipsilateral PFC, and there are other subcortical targets aside from the habenula and pontine nuclei. Following from this, why were the habenula and pontine nuclei targeted for these experiments?

We have now changed the terms to "contralateral PFC-projecting (cPFC-projecting)", "habenula-projecting" and "pons-projecting". Since a large population of layer-5 PCs projects to many subcortical targets and we are not able to make a complete survey of these cells, we only examined those with relatively long axons projecting to habenula and pontine nuclei.

The authors conclude that autapses persist during brain development into adulthood in the main manuscript. Therefore, the age of the mice should also be reported in the main manuscript.

The ages of the juvenile and adult mice have now been reported in the main text and the methods section.

It is not clear, from supplementary figure 1, that the recorded cell was indeed labelled by the beads.

Higher magnification images of the cell bodies has been added to the figure. Please see the figure below.

In figure 1a, where were the putative autaptic contacts located in the example cell?

We thank the reviewer for pointing this out. We have now added arrows pointing to the location of autaptic contacts.

How did the amplitudes of the aEPSCs compare between human and mouse?

Since it is very difficult to obtain dual somatic and axonal recording from adult human PCs, we are not able to obtain unitary aEPSCs from human PCs. In this revision, we compared the average peak amplitude (human: -15.3 ± 1.3 pA, $n = 4$; Mouse: -19.3 ± 2.6 pA, $n = 5$) and the kinetics (human: rise 1.09 ± 0.02 ms, decay 6.35 ± 0.32 ms; mouse: rise 0.91 ± 0.03 ms, decay 3.92 ± 0.40 ms) of desynchronized aEPSCs obtained from human and adult mouse PCs in Sr^{2+} ACSF.

Responses to Reviewer #3:

In this paper, Yin and colleagues examine the functional presence of autaptic excitatory responses in cortical pyramidal neurons. Whereas autaptic GABAergic transmission is prominent in a subset of cortical inhibitory cells, in the literature there is contradictory morphological evidence whether also glutamatergic PCs can express self-innervation. The authors set out to determine whether it is possible to record autaptic glutamatergic transmission from cortical principal neurons. The paper is divided into two blocks: results of Fig. 1-5 characterize autaptic responses in PCs and those of Fig. 6-7 examine a possible functional role for fast glutamatergic self-excitation of large PCs.

The manuscript reports several interesting findings: the authors demonstrate

the existence of autaptic glutamatergic neurotransmission in several specific PC subtypes; they reveal that autaptic transmission is strong, and they provide some insights on how self-innervation modulate firing. Results are interesting, of very high quality and overall compelling. Yet, the manuscript should be improved, as in the present form it suffers from several limitations.

Major:

1. The authors used a clever approach, by recording both from axonal blebs and somas coupled to 2-photon laser axotomy. Without the nuance of the large spike and membrane distortions due to voltage jumps, autaptic responses look very nice. However, it will be important to see if these responses obtained after axotomy are similar to those obtained with the intact axon. This is especially relevant for results of Fig. 4 in which autaptic and synaptic transmissions are compared. In this respect, it is somehow disappointing that for results in Fig. 2 the authors do not show actual autaptic responses obtained after capacitance cancellation and after subtracting traces before and after application of GluR antagonists. Inferring autaptic properties from Sr-dependent asynchronous release in response to AP trains can be indirect and problematic, as asynchronous autaptic quantal events will mix with spontaneous activity.

We thank the reviewer for pointing out this. Since it is not possible to obtain unitary aEPSCs before axotomy, we are not able to compare their waveforms before and after axotomy. However, we have now compared the peak amplitude and kinetics of spontaneous EPSCs (presumably arising from recurrent excitatory synapses) before and after axotomy. We found no significant change in the average peak amplitude (before: -17.5 ± 0.90 pA vs. after: -18.6 ± 0.80 pA, $n = 21$, $P = 0.29$, Wilcoxon signed-rank test. Please see the figure below), the rise time (1.10 ± 0.03 vs. 1.10 ± 0.04 ms, $P = 0.75$, Wilcoxon signed-rank test) and the decay time constant (5.28 ± 0.22 vs. 5.48 ± 0.29 ms, $P = 0.43$, Wilcoxon signed-rank test. New Supplementary Fig. 1). These results suggest that axotomy will not produce changes to autaptic responses. We have added these results to the revised manuscript.

In accordance to the reviewer's suggestion, we performed subtraction of the traces before and after the application of Kyn. The inset in panel-b (gray trace) is a result trace of subtraction (Control - Kyn), and that in panel-a is an expanded trace of the boxed region. We have now added this information to the new Fig. 2.

In the Sr²⁺ experiments that examined the properties of asynchronous aEPSCs, since the spontaneous EPSCs arose from recurrent synapses were very sparse (frequency: 0.69 ± 0.08 Hz, $n = 5$ cells), the probability of mixture of these spontaneous events with asynchronous autaptic quantal events in a time window less than 1 second was low.

2. The authors measure a response latency of ~ 2 ms, which is compatible with unitary connections between PCs, and which could perhaps allow showing autaptic EPSPs in current clamp. In this context, it is important that the authors should indeed measure how autaptic transmission changes spike waveform. The authors show two cell types with different AHP, but they did not directly test how autaptic transmission modify post-spike Vm. They should measure autaptic EPSP peaks, peak-times, if visible, and/or measure post-spike AHP and CV of Vm before and after application of glutamate receptor antagonists. This will also validate their results of Fig. 6.

We totally agree with the reviewer in this point. Since the autaptic EPSP overlaps with the large conductance changes during and after the action potential, the aEPSP peaks are not visible. In this revision, we performed additional recordings from layer-5 PCs showing autaptic responses (examined in Sr²⁺-ACSF after each experiments). In current clamp mode, we compared the AP waveforms before and after the application of Kyn. The half-width of APs showed no significant change after the application of 1.5 mM Kyn (from 0.64 ± 0.04 to 0.63 ± 0.04 ms, $n = 8$, $Z = 1.12$, $P = 0.26$, Wilcoxon signed-rank test). Consistent with the results showing in Figure 6, the post-spike AHP increased from 2.95 ± 0.39 to 4.88 ± 0.41 mV

($Z = -2.52$, $P = 0.011$, Wilcoxon signed-rank test). The CV of V_m 5 ms after the AP peak decreased from 0.24 ± 0.07 to 0.11 ± 0.06 mV, but without significance ($Z = 0.84$, $P = 0.40$, Wilcoxon signed-rank test).

3. Absence of NMDARs: this finding is quite interesting and highlights a clever mechanism for autaptic responses to prevent STDP plasticity, which is inherent with unavoidable pre- and postsynaptic timing. However, again, the method employed to examine AMPA and NMDA component of autaptic responses is quite indirect. Why didn't the authors examine the existence (or lack thereof) of NMDARs using autaptic responses obtained after axotomy? This would have allowed depolarizing the PCs to remove Mg block of NMDARs and/or applying glutamate receptor selective pharmacology on 'clean' and unequivocal autaptic responses. Importantly the use of CNQX is not ideal, as this drug acts on the glycine site of NMDARs. The authors used gly in their recordings, but still, more specific blockers for AMPARs are available. Also, analysis of NMDAR expression at synapses using miniature events might not be ideal, as it is impossible to decipher the origin of quantal events. I would suggest using unitary connections at PC-PC pairs and/or minimal stimulations of axons impinging distal layer 1 dendrites. These experiments will provide direct evidence for synapse vs. autapses specificity of NMDAR expression, which, again, is quite an interesting finding.

Very good suggestions from the reviewer. Indeed, it would be nice if we could examine the contribution of NMDAR to the unitary autaptic responses in the axotomy experiments. To do that, however, we encountered several technical difficulties. Since we perform dual soma and axon recording using normal internal solution (K^+ -based pipette solution), it is difficult to depolarize the somatic V_m to ~ 40 mV to remove Mg^{2+} block of the NMDARs. It would be better if we could perform somatic recording using QX314-containing Cs^+ -based pipette solution (with voltage-gated Na^+ and K^+ channels blocked) and axonal recording using normal pipette solution (ensure action potential generation in the axon). However, axonal AP generation would be disrupted before axotomy because of diffusion of QX314 and Cs^+ into the axonal compartments. Re-patching the soma with QX314 and Cs^+ internal solution after regular dual soma-axon recording and axotomy, the axon recording is often destroyed because of mechanical instability during re-patching. We also tried experiments in a bath with Mg^{2+} omitted but adding $100 \mu M$ D-serine or $10 \mu M$ Glycine (to enhance NMDAR activity), however, under these experimental conditions, slices generated noisy background synaptic activity and periodic epileptiform activity.

According to the reviewer's suggestion, we have now examined the effect of NBQX, an AMPA receptor antagonist (more specific than CNQX). We found that the application of $10 \mu M$ NBQX could completely block the

autaptic events (see the figure below). After drug application, the current noise level before and after the voltage pulse showed no significant difference (1.49 ± 0.21 vs. 1.61 ± 0.17 pA, $n = 6$, $t_5 = -1.56$, $P = 0.18$, paired Student's t -test).

We also followed the suggestions from the reviewer and delivered extracellular stimulation to evoke EPSCs in the recorded layer-5 PC. Since most of the recurrent excitatory synapses target the basal dendrites (similar to autapses), we chose to deliver minimal stimulation to the neighboring tissue at layer 5, instead of layer 1. We performed these experiments in a similar condition as aEPSCs in Sr²⁺ ACSF with Mg²⁺ omitted but addition of glycine. We found that the decay time constant decreased after the application of 50 μM APV (Control: 3.76 ± 0.17 vs. APV: 2.74 ± 0.31 ms, $n = 7$, $t_6 = 3.55$, $P = 0.01$, paired Student's t -test). The rise time also decreased from 0.69 ± 0.05 to 0.54 ± 0.05 ms ($n = 7$, $t_6 = 3.63$, $P = 0.01$, paired Student's t -test). The peak amplitude showed no significant change (-18.9 ± 2.30 vs. -15.9 ± 1.30 pA, $n = 7$, $Z = 1.01$, $P = 0.38$, Wilcoxon signed-rank test). These results are consistent with previous findings showing NMDA component in recurrent excitatory synapses in the cortex (Markram, 1997; Markram et al., 1997; Wang et al., 2008). Please see the figure showing below.

4. *The experiments of Fig. 6f-g are not entirely convincing. It is not clear why the authors need to induce spikes with extracellular stimulation, which in the neocortex, might induce unwanted polysynaptic activity and thus complicate the interpretation of the results. Moreover, intracellular BAPTA diffusion might very likely induce effects on AHP peak and duration. Finally, the differential effect between short- and long-axon PCs (the differences between red and blue data in Fig. 6g) does not seem to be that striking (compare the last points of the time-course). Here, I suggest a possible solution. Autaptic transmission is estimated to be quite strong: therefore, the ability to modulate burst duration can be directly estimated by repetitively inducing a single spike with a very short membrane depolarization and see if autaptic transmission can induce doublets or triplets. This can be tested by plotting the probability of obtaining a second spike in the absence and presence of GluR antagonists. If a second spike is common, in addition to modulate its probability, autaptic blockade will modulate its latency, as shown in Fig. 6a-c.*

In the experiments shown in Fig. 6f-g, we performed whole-cell recording from layer-5 PCs and delivered single electric shocks to the border of layer 6 and white matter. From our experience of studying the network activities (such as Up state and Down state) in modified ACSF (in mM: 1 Mg, 1 Ca and 3.5 K)(Hasenstaub et al., 2005; Shu et al., 2003a; Shu et al., 2003b), polysynaptic activity is minimized with single brief shocks in cortical slices maintained in normal ACSF (2 Mg, 2 Ca, 2.5 K). Because the only factor changed during recording was the diffusion of BAPTA into the recorded cell, we believe that the changes in intra-burst frequency reflect the effect of BAPTA blockade of autaptic transmission. Indeed, intracellular BAPTA would have an effect on after-burst AHP. Previous studies showed that intracellular BAPTA would reduce the slow after-burst hyperpolarization but exert no effect on intrinsic bursting in hippocampal pyramidal cells.

[Redacted]

We totally agree with the reviewer and appreciate the suggestion of experiments that directly investigate the role of autapse in regulating burst firing. Fig. 7a illustrates the experiment protocol: we repetitively stimulate the recorded cell with paired current pulses (1 ms in duration, 20 ms in inter-stimulus interval), the first pulse induces APs reliably, we varied the current of the second pulse. Exactly as suggested by the reviewer, we plotted the probability of obtaining a second spike in the absence and presence of GluR antagonist, Kyn. We found that blocking GluR reduced the probability of the second AP (Fig. 7b). In addition, we also applied cyclothiazide (CTZ, 100 μ M), a blocker of AMPA receptor desensitization, to enhance aEPSCs and found that the probability of the

second AP was dramatically increased (Fig. 7b). Since single brief current pulses could not evoke doublets and triplets (i.e. a second spike is uncommon in PFC PCs examined in our recordings), we were not able to examine the inter-spike interval within bursts. Please note that after each recording the recorded PC was bathed in Sr^{2+} -ACSF to examine whether it possessed autaptic connections (by showing desynchronized autaptic events as in Fig. 2). In the original submission, we only described these experiments as an examination of cell responsiveness to an immediate stimulation after AP generation. In this revision, we have now clarified this in main text.

Minor:

1. Results, line 102: “Compared to regular synaptic currents, these events were giant, with an average peak ...”. This is a bit misleading at this point in the manuscript, as the comparison is made much later.

Corrected.

2. Results, line 110-111: “The short onset latencies of these aEPSCs ... agree well with monosynaptic transmission.” This sentence requires a reference. I would suggest using Markram ... Sakmann J. *Physiol.* 1997, PMID: 9147328. This paper report an in-depth analysis of deep-layer cortical PC-PC connections.

We have now added this citation to the revised manuscript.

3. Line 303: “in intact tissue”. Acute brain slices cannot be defined as intact. I would perhaps write *ex vivo* or in acute brain slices keeping a good degree of intact synaptic connectivity.

Corrected.

References

Franceschetti, S., Guatteo, E., Panzica, F., Sancini, G., Wanke, E., and Avanzini, G. (1995).

Ionic mechanisms underlying burst firing in pyramidal neurons: intracellular study in rat sensorimotor cortex. *Brain research* 696, 127-139.

Hasenstaub, A., Shu, Y., Haider, B., Kraushaar, U., Duque, A., and McCormick, D.A. (2005). Inhibitory postsynaptic potentials carry synchronized frequency information in active cortical networks. *Neuron* 47, 423-435.

Kalisman, N., Silberberg, G., and Markram, H. (2005). The neocortical microcircuit as a tabula rasa. *Proc Natl Acad Sci USA* 102, 880-885.

Kole, M.H. (2011). First node of Ranvier facilitates high-frequency burst encoding. *Neuron* 71, 671-682.

Mantegazza, M., Franceschetti, S., and Avanzini, G. (1998). Anemone toxin (ATX II)-induced increase in persistent sodium current: effects on the firing properties of rat neocortical pyramidal neurones. *The Journal of physiology* 507 (Pt 1), 105-116.

Markram, H. (1997). A network of tufted layer 5 pyramidal neurons. *Cereb Cortex* 7, 523-533.

Markram, H., Lübke, J., Frotscher, M., Roth, A., and Sakmann, B. (1997). Physiology and anatomy of synaptic connections between thick tufted pyramidal neurones in the developing rat neocortex. *J Physiol* 500 (Pt 2), 409-440.

Shu, Y., Hasenstaub, A., Badoual, M., Bal, T., and McCormick, D.A. (2003a). Barrages of synaptic activity control the gain and sensitivity of cortical neurons. *The Journal of neuroscience : the official journal of the Society for Neuroscience* 23, 10388-10401.

Shu, Y., Hasenstaub, A., and McCormick, D.A. (2003b). Turning on and off recurrent balanced

cortical activity. *Nature* 423, 288-293.

Wang, H., Stradtman, G.G., 3rd, Wang, X.J., and Gao, W.J. (2008). A specialized NMDA receptor function in layer 5 recurrent microcircuitry of the adult rat prefrontal cortex. *Proc Natl Acad Sci USA* 105, 16791-16796.

Williams, S.R., and Stuart, G.J. (1999). Mechanisms and consequences of action potential burst firing in rat neocortical pyramidal neurons. *J Physiol* 521 Pt 2, 467-482.

Reviewers' Comments:

Reviewer #1:

Remarks to the Author:

The authors have adequately addresses my concerns raised on the original submission. I congratulate the authors on a very mice study.

Reviewer #2:

Remarks to the Author:

This manuscript by Yin et al has been greatly improved from the initial submission. The authors addressed all of my questions (and those of the other reviewers) and consequently the paper has been significantly improved.

Reviewer #3:

Remarks to the Author:

The authors have revised their manuscript with a new series of analyses and some more experiments. Overall, I think that the paper has improved. However, I feel that my general concerns were not fully addressed.

In particular, the response of my original major point 1 is a bit disappointing. My concern was whether autaptic responses are similar in the presence and absence of axotomy, but the authors show that sEPSCs do not change before and after cutting the axon. This is not surprising, since sEPSCs in slice are mostly AP-independent (spontaneous events are dominated by minis) and originate from distinct neurons. So, I cannot see how comparing sEPSCs before and after axotomy would answer the question.

Again, it is a pity that the authors could not isolate aEPSCs and aEPSPs with no axotomy and in the absence of Sr. In fact, subtracting the Kyn traces in Sr is not very informative, as glu-mediated events occur far away from the action currents. It would have been more informative to subtract the traces obtained with a single action current before and after applying GluR antagonists, and in the absence of Sr. This would have allowed better understanding: 1) whether axotomy does or does not alter synchronous autaptic transmission; 2) how autaptic responses differ from uEPSCs in pairs of PCs; 3) how autaptic potentials modulate AHP waveform. In other words, why couldn't the authors record autaptic responses in control (no Sr) conditions? The authors should clearly explain why autaptic currents and potentials could not be revealed upon trace subtraction, as it is commonly done for autaptic transmission from GABAergic interneurons or from PCs in culture.

Regarding the analysis of PC spikes and AHP in the presence and absence of Kyn. The authors measure changes in the AHP peak, and they obtain results that are in agreement with those shown in Fig. 6, but I could not find these measurements in the revised manuscript. Why? I think showing that spike waveform is modulated by autaptic transmission is important.

I have another concern regarding the results of Fig. 6a,b: The authors measure a wide range of EPSC amplitudes with a max value of 432 pA, which corresponds to ~6 nS. The average autaptic response is of ~150 pA, which corresponds to ~2nS conductance. Yet, in their dynamic clamp experiments of Fig. 6a,b they use conductance values, which are either not common (4 nS) or outside their max recorded values (8 nS). In particular, I am a bit worried of the data reported in Fig. 6b, right panel, as it refers to responses to an unrealistic autaptic conductance of 8 nS. Could the author explain why they used these conductance levels? What results would they obtain if they use values that are more realistic? This somewhat applies to the results of Fig. 7c.

NMDAR absence at autaptic contacts: I appreciate the authors' response. I understand the technical problems associated to test directly the presence or absence of NMDARs using a Cs- and QX-based intracellular solution. They show interesting results with minimal stimulation indicating NMDAR expression at glutamatergic synapses. Why didn't the authors include these results in the revised manuscript (either to complement Fig. 5 or in a Suppl. Figure)?

I would not use the term 'gigantic' for autaptic responses (Results and Discussion).

Reviewers' comments:

Reviewer #1 (Remarks to the Author):

The authors have adequately addresses my concerns raised on the original submission. I congratulate the authors on a very nice study.

Reviewer #2 (Remarks to the Author):

This manuscript by Yin et al has been greatly improved from the initial submission. The authors addressed all of my questions (and those of the other reviewers) and consequently the paper has been significantly improved.

Reviewer #3 (Remarks to the Author):

The authors have revised their manuscript with a new series of analyses and some more experiments. Overall, I think that the paper has improved. However, I feel that my general concerns were not fully addressed.

In particular, the response of my original major point 1 is a bit disappointing. My concern was whether autaptic responses are similar in the presence and absence of axotomy, but the authors show that sEPSCs do not change before and after cutting the axon. This is not surprising, since sEPSCs in slice are mostly AP-independent (spontaneous events are dominated by minis) and originate from distinct neurons. So, I cannot see how comparing sEPSCs before and after axotomy would answer the question.

Since both autaptic and recurrent synaptic contacts form onto the somatodendritic compartments, axotomy may produce similar changes in the waveform of aEPSC and sEPSC. Because there is no significant change in sEPSCs after axotomy, we speculate that aEPSC waveforms should be similar before and after axotomy.

[Redacted]

Again, it is a pity that the authors could not isolate aEPSCs and aEPSPs with no axotomy and in the absence of Sr. In fact, subtracting the Kyn traces in Sr is not very informative, as glu-mediated events occur far away from the action currents. It would have been more

informative to subtract the traces obtained with a single action current before and after applying GluR antagonists, and in the absence of Sr. This would have allowed better understanding: 1) whether axotomy does or does not alter synchronous autaptic transmission; 2) how autaptic responses differ from uEPSCs in pairs of PCs; 3) how autaptic potentials modulate AHP waveform. In other words, why couldn't the authors record autaptic responses in control (no Sr) conditions? The authors should clearly explain why autaptic currents and potentials could not be revealed upon trace subtraction, as it is commonly done for autaptic transmission from GABAergic interneurons or from PCs in culture.

[Redacted]

[Redacted]

Regarding the analysis of PC spikes and AHP in the presence and absence of Kyn. The authors measure changes in the AHP peak, and they obtain results that are in agreement with those shown in Fig. 6, but I could not find these measurements in the revised manuscript. Why? I think showing that spike waveform is modulated by autaptic transmission is important.

Thanks to the reviewer for pointing this out. We performed these analysis and provided in the responses to reviewers but forgot to include these data in the main text. We have now added these information to the revised manuscript (line 274-280): “In autaptic PCs, we compared the AP waveforms before and after the application of Kyn (1.5 mM). The half-width of APs showed no significant change (from 0.64 ± 0.04 to 0.63 ± 0.04 ms, $n = 8$, $Z = 1.12$, $P = 0.26$, Wilcoxon signed-rank test). The post-spike AHP increased from 2.95 ± 0.39 to 4.88 ± 0.41 mV ($Z = -2.52$, $P = 0.011$, Wilcoxon signed-rank test). The CV of V_m 5 ms after the AP peak decreased from 0.24 ± 0.07 to 0.11 ± 0.06 mV, but without significance ($Z = 0.84$, $P = 0.40$, Wilcoxon signed-rank test).”

I have another concern regarding the results of Fig. 6a,b: The authors measure a wide range of EPSC amplitudes with a max value of 432 pA, which corresponds to ~6 nS. The average autaptic response is of ~150 pA, which corresponds to ~2nS conductance. Yet, in their dynamic clamp experiments of Fig. 6a,b they use conductance values, which are either not common (4 nS) or outside their max recorded values (8 nS). In particular, I am a

bit worried of the data reported in Fig. 6b, right panel, as it refers to responses to an unrealistic autaptic conductance of 8 nS. Could the author explain why they used these conductance levels? What results would they obtain if they use values that are more realistic? This somewhat applies to the results of Fig. 7c.

In this study, we used acute slice preparation. As stated in the manuscript, slicing of the brain would cut and reduce the complexity of dendrites and axons, and thus cause underestimation of the number of autaptic contacts. We speculate that the actual autaptic responses in the brain should be even greater than the recorded ones in slices. Therefore, in our dynamic clamp experiments, we examined a range of autaptic conductances (g_{aut} : 0, 2, 4, 8, 10 nS, Fig. 7b left panel) including that (8 nS) slightly greater than the max value (6.2 nS). As mentioned by the reviewer, 2 nS corresponds to the average autaptic response (~150 pA, driving force: 70 mV). 4 nS is within the range of measured conductances (< 6 nS). In Fig. 7c, we used an autaptic conductance of 4 nS, which is within the measured conductance range. In this revision, we have added a plot for 4 nS to Fig. 6b (right panel, please see below).

NMDAR absence at autaptic contacts: I appreciate the authors' response. I understand the technical problems associated to test directly the presence or absence of NMDARs using a Cs- and QX-based intracellular solution. They show interesting results with minimal stimulation indicating NMDAR expression at glutamatergic synapses. Why didn't the authors include these results in the revised manuscript (either to complement Fig. 5 or in a Suppl. Figure)?

We thought that it could be enough to provide changes in kinetics of desynchronized evoked EPSCs after the application of APV and cite previous studies showing NMDAR component in recurrent glutamatergic synapses. But we agree with the reviewer, we need to show this nice experiment in the manuscript. Please see the new Supplementary Figure 4.

I would not use the term 'gigantic' for autaptic responses (Results and Discussion).

We have now changed it to "large".

Reviewers' Comments:

Reviewer #3:

Remarks to the Author:

The paper has significantly improved and I do not have major concerns. It is a pity that the authors cannot include the aEPSC subtraction analysis obtained with J. Huguenard's method, as it looks convincing...

Regarding the size of autaptic conductance used in the dynamic clamp experiments, the authors included a new set of data obtained with 4 nS. They should nonetheless discuss this point in the paper, as the average autaptic response they measure is ~2 nS. The authors should justify their choice of using high levels of conductance in their dynamic clamp experiments.

To justify the use of very high autaptic conductance values (>6 nS), the authors write: "As stated in the manuscript, slicing of the brain would cut and reduce the complexity of dendrites and axons, and thus cause underestimation of the number of autaptic contacts". I don't find this argument particularly convincing, as there is no evidence that the strength of local synaptic connections is higher in in vivo preparations. One has to work with what he/she measures, and not what could be speculated as happening in an intact brain. I would not use this argument in the Discussion.

REVIEWERS' COMMENTS (in italic):

Reviewer #3 (Remarks to the Author):

[Redacted]

Regarding the size of autaptic conductance used in the dynamic clamp experiments, the authors included a new set of data obtained with 4 nS. They should nonetheless discuss this point in the paper, as the average autaptic response they measure is ~2 nS. The authors should justify their choice of using high levels of conductance in their dynamic clamp experiments.

We agree with the reviewer in this point. We should have clarified this in previous revision and provided the maximum peak amplitude of single trial aEPSCs (not just the maximum averaged amplitude). Please see the Results section in the manuscript (Line 167-170): “We found that aEPSCs were large with an average peak amplitude of -149 ± 11 pA across cell population ($n = 62$ cells, the average for individual cells ranged from -27 to -432 pA, single trials could reach up to -524 pA).” We have also clarified the selection of conductance values in the Results section (Line 261-266): “In axotomy experiments, the average aEPSC amplitude across cell population (-149 pA) and the maximum amplitude among individual trials (-524 pA) correspond to 2.1 and 7.5 nS with a driving force of 70 mV, respectively. Thus, we examined a range of g_{aut} including conductances (2 and 4 nS) within the range of measured values, as well as those (8 and 10 nS) slightly higher than the maximum value.”

To justify the use of very high autaptic conductance values (>6 nS), the authors write: "As stated in the manuscript, slicing of the brain would cut and reduce the complexity of dendrites and axons, and thus cause underestimation of the number of autaptic contacts". I don't find this argument particularly convincing, as there is no evidence that the strength of local synaptic connections is higher in in vivo preparations. One has to work with what he/she measures, and not what could be speculated as happening in an intact brain. I would not use this argument in the Discussion.

Considering the maximum aEPSC amplitude (-524 pA) among single trials corresponds to 7.5 nS, we think that the selected g_{aut} of 8 and 10 nS are only slightly greater than the maximum conductance measured. We have now clarified this in the Result section. In Discussion, we do not use this argument about reduced dendrite and axon complexity to explain the use of high levels of conductances.